# Phase diagram and conformal string excitations of square ice using gauge invariant matrix product states

**Ferdinand Tschirsich[1*], Simone Montangero[1,2,3] and Marcello Dalmonte[4,5]**

**1** Institute for Complex Quantum Systems and Center for Integrated Quantum Science and Technologies, Universität Ulm, 89069 Ulm, Germany
**2** Dipartimento di Fisica e Astronomia, Universitá degli Studi di Padova, 35131 Padova, Italy
**3** Theoretische Physik, Universität des Saarlandes, 66123 Saarbrücken, Germany
**4** The Abdus Salam International Centre for Theoretical Physics, 34151 Trieste, Italy
**5** SISSA, 34150 Trieste, Italy

⋆ ferdinand.tschirsich@uni-ulm.de

## Abstract

We investigate the ground state phase diagram of square ice — a $U(1)$ lattice gauge theory in two spatial dimensions — using gauge invariant tensor network techniques. By correlation function, Wilson loop, and entanglement diagnostics, we characterize its phases and the transitions between them, finding good agreement with previous studies. We study the entanglement properties of string excitations on top of the ground state, and provide direct evidence of the fact that the latter are described by a conformal field theory. Our results pave the way to the application of tensor network methods to confining, two-dimensional lattice gauge theories, to investigate their phase diagrams and low-lying excitations.



# 1  Introduction

Since the formulation of the density-matrix renormalization group algorithm by White in 1992 [1], numerical techniques based on tensor network (TN) ansätze have found widespread application in condensed matter theory [2,3]. While initially restricted to low-lying excitations of one-dimensional (1D) systems, these techniques are nowadays routinely used to investigate real-time dynamics as well as finite temperature. Over the last five years, a flurry of activity has been devoted to show how TN methods can be extended to lattice gauge theories (LGTs) [4–22] — i.e., to theories including dynamical gauge fields. Numerical simulations based on gauge invariant tensor networks (GITNs) have been applied to several 1D Abelian and non-Abelian LGTs, to investigate phenomena as diverse as, e.g., string-breaking and low-lying spectra (see, e.g., Ref. [23, 24]). The vision behind this programme is to ultimately extend these methods to higher-dimensional LGTs displaying confinement, and thus render numerical regimes accessible, which are typically not tractable with Monte Carlo (MC) techniques. These include finite chemical potentials and real-time dynamics — describing, for instance, the time-evolution of many-body systems governed by quantum-chromodynamics [25, 26].

    In this work, we apply GITNs to a *two-dimensional, confining* LGT — a U(1) quantum link model also known as square ice [27–30]. This model represents an ideal testing ground for exploring possibilities and limitations of GITNs in 2D. Its phase diagram is rich, presenting three distinct confined phases: a Néel phase, a resonating valence bond solid (RVBS) phase, and a columnar phase. From the simulation perspective, its dynamics is solvable with Monte Carlo simulations, which allows us to qualitatively and quantitatively benchmark 2D GITN approaches. Still, it is worth mentioning that an efficient Monte Carlo algorithm for this model has been devised only very recently [29], indicating highly nontrivial quantum dynamics at low energies.

    Our purpose here is twofold. The first objective is to validate and assess the performances of GITNs in investigating 2D lattice gauge theories which display confinement (for deconfined theories, see Ref. [4]). The second one is to exploit the predictive power of these techniques, and in particular, the possibility of gaining novel insights based on entanglement properties — e.g., entanglement spectra and entropies — of the system.

    In order to address the aforementioned objectives, we carry out extensive GITN simulations

using the time-evolving block decimation (TEBD) algorithm [31, 32] for various cylinder geometries including up to $\simeq 600$ spins. We start our analysis with a focus on expectation values and correlations of local observables, which allow us to clearly distinguish the different phases of the model, and to locate the transition point between Néel and RVBS phases in agreement with alternative studies based on exact diagonalization. Following this, we specifically address aspects related to confinement by calculating Wilson loop expectation values, and string tensions by introducing a pair of static charges. Our results clearly show that the string tension is finite at the Néel–RVBS transition, in agreement with recent MC simulations. This first part of the analysis shows that GITNs compare well to other methods in the zero-density regime, even if we are not able to reach sizes as large as the ones accessible to MC.

In the second part, we revisit the phase diagram of square ice from an entanglement perspective, allowed by the fact that GITNs provide direct access to the system wave-function of both the ground state and low-lying excited states. We show how the entanglement spectrum — the spectrum of the reduced density matrix obtained by tracing out a part of the degrees of freedom from the ground state wave function — displays distinctive features in different confined phases, reflecting the corresponding symmetry breaking patterns. Then, we carry out a detailed study of string excitations in the system from an entanglement perspective. In particular, we provide entanglement-based evidence for the fact that such string excitations are described by a conformal field theory with central charge $c = 1$ — effectively behaving as a compactified boson. This approach has never been applied to LGTs, and, compared to other methods based on energy spectroscopy of the Lüscher term [33], it allows us to estimate the central charge from moderate system size simulations.

The paper is organized as follows. In Sec. 2, we introduce the model Hamiltonian, summarize the main features reported in the literature about the phase diagram, and provide details on our simulation strategy, based on imaginary time-evolution by means of TEBD on gauge-invariant matrix-product states (MPS). In Sec. 3, we present our results on order parameters, correlation functions, and Wilson loops, followed by a numerical analysis of entanglement properties and string excitations in the RVBS phase in Sec. 4. We conclude in Sec. 5.

## 2 Model Hamiltonian and Methods

### 2.1 Square Ice Hamiltonian

The Hilbert space of the square ice model is defined — analogously to conventional lattice gauge theories — by a set of spin-1/2 degrees of freedom that reside on the edges, or *links*, $\mu$ of a square lattice.

The system Hamiltonian reads [28, 29, 34, 35]:

$$H = \sum_{\square}\left(-f_{\square} + \lambda f_{\square}^2\right), \tag{1}$$

where the summation goes over all square plaquettes, and the plaquette operator $f_{\square} = \sigma_{\mu_1}^{+}\sigma_{\mu_2}^{+}\sigma_{\mu_3}^{-}\sigma_{\mu_4}^{-} + \text{H.c.}$ inverts the spins on the respective boundary links $\mu_1, \dots, \mu_4$ of oriented plaquettes, as shown in Fig. 1(a). The first term in the Hamiltonian is the equivalent of a magnetic field term, and the second term with coupling parameter $\lambda$ — also known as Rokhsar-Kivelson (RK) term [36] — imposes a finite energy on oriented plaquettes (i.e. on two out of four configurations where the product of all spin operators in the $z$-basis is $+1$).

It is convenient to represent spin up (down) by right- and upwards (left- and downwards) pointing arrows on horizontal and vertical links, respectively. In this picture, $f_{\square}$ flips oriented plaquettes and projects out anything else, while $f_{\square}^2$ counts oriented plaquettes.

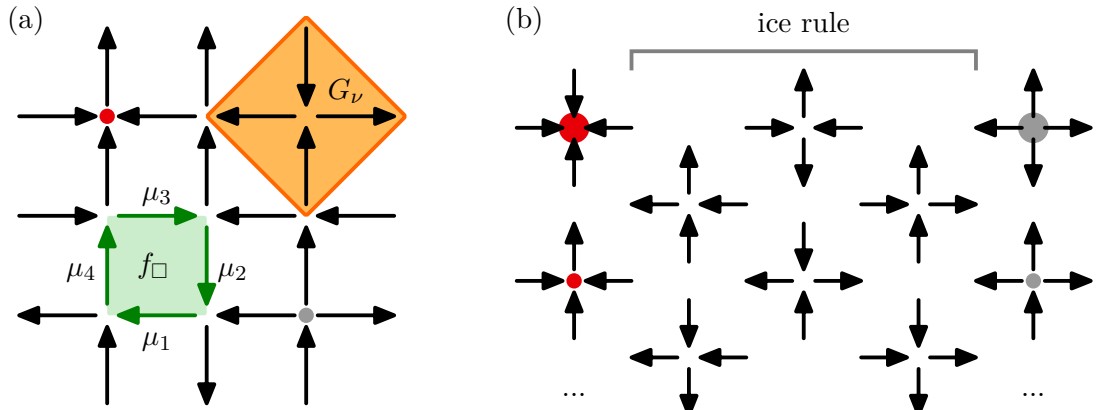

Figure 1: Square ice quantum link model. Arrows visualize spin-1/2 polarizations on the links; small (large) red and grey dots indicate positive and negative static charges $\bar{c} = \pm 1 (\pm 2)$. (a) Lattice configuration with two charges $\bar{c} = \pm 1$ and one oriented plaquette (green) which can be flipped by $f_\square$ from counter- to clockwise orientation. The U(1) gauge symmetry generator $G_\nu$ counts in- minus outbound arrows at vertices (orange). (b) Vertex configurations: The number of in- minus outbound arrows fixes the charge. For $\bar{c} = 0$, six 'ice rule' configurations (2 in, 2 out) are possible.

We study the system on finite cylindrical lattices, and label the vertices $\nu = n\hat{x} + m\hat{y} = (n, m)$ with $n \in \{1, \ldots, L_x\}$ and $m \in \{1, \ldots, L_y\}$. Therein, we locate the spins on centres of horizontal and vertical links, $\mu = (n + 1/2, m)$ and $(n, m + 1/2)$, respectively. Spins on the open boundary at $n = 0$ and $n = L_x$ are fixed, and periodic boundary conditions apply in the $\hat{y}$-direction with $m \equiv m + L_y$.

The Hamiltonian (1) is invariant under a local U(1) gauge symmetry generated by the vertex operator

$$G_\nu = \sum_{\hat{i} \in \{\hat{x}, \hat{y}\}} \left( \sigma^z_{\nu - \hat{i}/2} - \sigma^z_{\nu + \hat{i}/2} \right), \tag{2}$$

where $\hat{x}, \hat{y}$ are unit lattice vectors along either dimension (see Fig. 1(a)). $G_\nu$ counts the difference between in- and outbound arrows at any vertex $\nu$, and the dynamics of the square ice decouples into sectors with different sets of background charges $\bar{c}_\nu$ as stated in the Gauss-law

$$G_\nu \equiv \bar{c}_\nu \in \{0, \pm 1, \pm 2\}. \tag{3}$$

We primarily operate in absence of charges $G_\nu \equiv 0$, where every vertex joins exactly two in- and two outbound arrows, a condition termed 'ice rule' for its analogy to the hydrogen configuration in hexagonal water ice [37, 38]. The ice-rule configuration space is equivalent to that of the six-vertex model (see Fig. 1(b)), and its dimension grows as $W^{L \times L}$ with Lieb's square-ice constant $W = (4/3)^{3/2} \approx 1.54$ on periodic $L \times L$ square lattices [39, 40]. This can be compared to $W = 4$ when ice rules are absent.

The gauge invariant subspace states can be identified with the low-energy manifold of the spin-1/2 frustrated antiferromagnetic XXZ model on a lattice of corner-sharing cross-linked squares (the two-dimensional analogue of a pyrochlore lattice), in the limit of strong anisotropy $J_{xy} \ll J_z$ [28] (an equivalent mapping can be obtained in Ising models, see Ref. [27]). The plaquette flip operator $f_\square$ arises from perturbative tunnelling elements between ice-states at the order of $J^2_{xy}/J_z$, on top of which a tunable chemical potential on oriented plaquettes is added to arrive at Eq. (1). In this framework, spin-excitations manifest

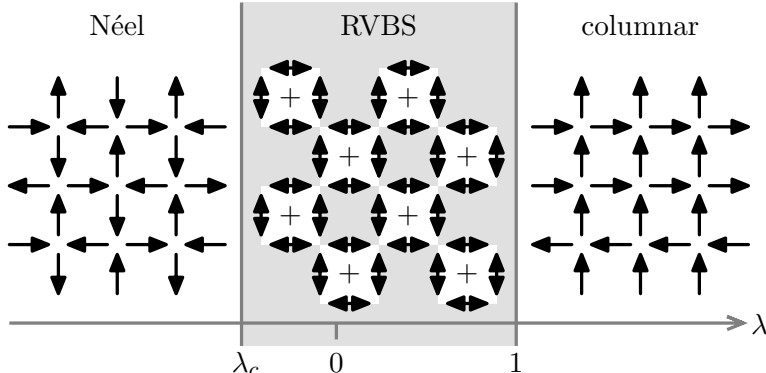

Figure 2: Zero temperature phase diagram as a function of the Rokhsar-Kivelson coupling $\lambda$, depicted with cartoon lattice configurations. A quantum phase transition at $\lambda_c$ and the RK-point $\lambda_{\text{RK}} = 1$ separate a resonating valence bond solid (RVBS) phase (shaded) from Néel- and columnar phases.

in the form of nonzero charges and are separated by a gap in the order of $J_z$. Analytical and numerical results on the dynamics of various excitations are established in [28, 41].

The square ice of Eq. (1) changes into a model of densely packed dimers [36], when instead of enforcing the ice rule, one places alternating charges $\bar{c}_{(n,m)} = (-1)^{n+m}$ at each vertex [29]. As in the dimer model case [36], one can introduce winding numbers $w_x$ and $w_y$ that determine the flux (difference from in- and outbound arrows) through a vertical or horizontal cut of the lattice. On the cylinder, we define $w_y$ as the eigenvalue of $\sum_n \sigma^z_{(n,1/2)}$, and $w_x$ is fixed due to the open boundary conditions. As a consequence of gauge-invariance, these winding numbers are constants of motion and each pair of winding numbers constitutes a decoupled sector of the dynamics. Further symmetries of Hamiltonian (1) are global translation- and charge-conjugation invariances, the latter of which inverts all spin and negates any static charge placed on the lattice (see also Appendix B).

## 2.2 Zero Temperature Phase Diagram

The phase diagram of the square ice has been explored from numerical and analytical arguments [28, 29]. At zero temperature, it comprises a RVBS phase, sandwiched by a Néel phase and a columnar phase (see Fig. 2).

For $\lambda \to -\infty$, the number of oriented plaquettes is maximized in the ground state. Long range Néel ordered spin-correlations spontaneously break translational symmetry. Charge conjugation, which leaves the number of oriented plaquettes invariant, is broken as well, and the existence of degenerate symmetric and antisymmetric Néel ground states indicates a Néel phase that extends for $\lambda \leq \lambda_c$.

At $\lambda = \lambda_c$, the Néel phase terminates in a weakly first order quantum phase transition, pinpointed by an energy level crossing of the antisymmetric Néel-state with an other excited state. From extrapolating exact diagonalization of periodic lattices including up to 64 spins, the transition has been estimated to occur in the thermodynamic limit at $\lambda_c \approx -0.3727$ [28]. Further global fits to spectral gaps, including higher excited zero-momentum states, lead to $\lambda_c \approx -0.359(5)$ from up to $6 \times 6$ lattices [29].

For $\lambda_c < \lambda < \lambda_{\text{RK}}$, a phase of resonating plaquettes maximizes the kinetic energy in resonating between the two flipped orientations of individual plaquettes. Since resonating plaquettes repel each other onto one of two possible separate sublattices, translational symmetry remains broken, but charge conjugation is restored with two conjugation-invariant ground states. The

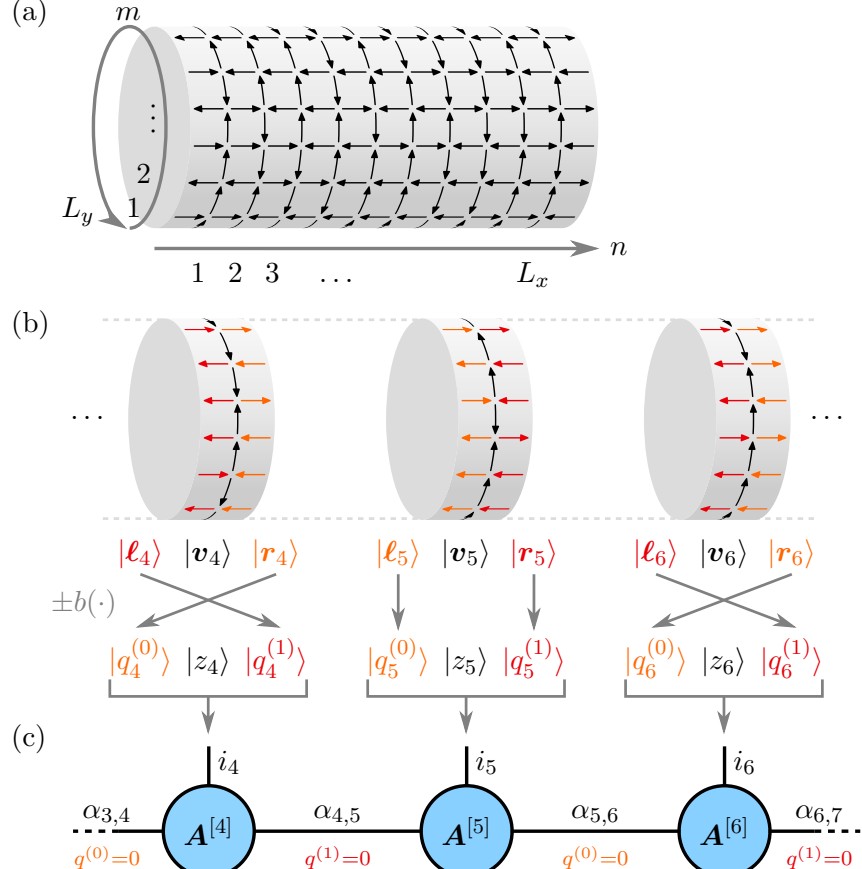

Figure 3: Mapping to MPS. (a) Cylindrical lattice. (b) Spins are grouped by computational site $n$, and encoded in U(1) quantum numbers $q_n^{(k)}$ and $z_n$. (c) Corresponding MPS segment with site-indices $i_n \sim (q_n^{(0)}, q_n^{(1)}, z_n)$ (open lines) and bond-indices $\alpha_{n,n+1}$ (shared lines). The full quantum state is encoded in the symmetry invariant tensors $\boldsymbol{A}^{[n]}$ (blue). A gauge invariant state has fixed quantum numbers $q^{(k)} = 0$ on odd-even ($k = 0$) and even-odd ($k = 1$) bonds.

RVBS phase includes the $\lambda = 0$ point, and extends up to the RK-point $\lambda_{\text{RK}} = 1$, characterized by a vanishing string tension and a ground state wave function which can be written as an equal weight superposition of closed loops [27].

Finally, for $\lambda > \lambda_c$, all configurations without any oriented plaquettes become ground states of the square ice. Those *columnar* configurations are characterized by long-ranged parallel alignment of spins along vertical or horizontal lines of links, with either all horizontal or all vertical lines locked in parallel alignment.

The nature of spinon-excitations (i.e. nonzero charges) is known to be deconfining in the columnar phase, including the RK-point $\lambda_{\text{RK}}$ [28]; and confining for $\lambda < \lambda_{\text{RK}}$, including the weakly-first-order transition point $\lambda_c$, where Monte Carlo simulations on large systems (up to $180 \times 180$) have revealed a small but finite string tension between static charges [29]. The fact that confinement is the only possible scenario here is related to the continuum-limit behaviour of the theory (recovered via dimensional reduction in quantum link models, see Ref. [42]), where confinement is due to monopole contributions [43].

## 2.3 Mapping to Gauge-Invariant Matrix Product States

In order to approximate the ground state of the square ice with matrix product states (MPSs) [44], we map the cylindrical system onto an open chain of length $L_x$ by combining spins with the same $\hat{x}$-coordinates into sites $n = 1, \ldots, L_x$. Starting from a standard tensor-product basis, spanned by the local $z$-basis states $\sigma_\mu^z |s_\mu\rangle = s_\mu |s_\mu\rangle$, $s_\mu \in \{\pm 1/2\}$, we merge all spins from vertical links into vectors

$$|\boldsymbol{v}_n\rangle := \bigotimes_m |s_{(n,m+1/2)}\rangle. \tag{4}$$

Spins on the adjacent horizontal links to the 'left' and 'right' of vertices with $\hat{x}$-coordinate equal to $n$ can be collectively addressed in vectors $|\boldsymbol{\ell}_n\rangle := \bigotimes_m |s_{(n-1/2,m)}\rangle$ and $|\boldsymbol{r}_n\rangle := \bigotimes_m |s_{(n+1/2,m)}\rangle$, respectively. The process is visualized in Fig. 3. We later fix specific boundary conditions $\boldsymbol{\ell}_1$ and $\boldsymbol{r}_{L_x}$ (see Appendix B).

A local computational basis for the MPS is constituted by states $|i_n\rangle \in \{|\boldsymbol{\ell}_n\rangle |\boldsymbol{v}_n\rangle |\boldsymbol{r}_n\rangle\}$, from which we remove all configurations that do not fullfill the Gauss-law Eq. (3) at each vertex $(n, m)$, $m \in \{1, \ldots, L_y\}$. This choice of basis immediately restricts our computations to a physical gauge sector, and is a very general recipe for tensor network simulations of lattice gauge symmetries [5, 45]. However, it comes at the cost of doubly-counted horizontal spin degrees of freedom (DOFs). This redundancy must be compensated by imposing the constraints

$$\boldsymbol{\ell}_{n+1} = \boldsymbol{r}_n \tag{5}$$

on our MPS representation, for instance by means of projection or energy penalties.

Here we follow a more direct approach, which employs the same numerical structures that encode global Abelian symmetries [46], and effectively extends their field of application to lattice gauge symmetries, including non-Abelian gauge symmetries [12]. The idea is described in more general terms in Appendix A, while this section focuses on the specific gauge-invariant MPS used for the results of this paper. It is highly efficient in that it entirely eliminates the redundant, unphysical DOFs by encoding the MPS matrices in form of U(1) symmetry invariant tensors [46–48]. In more detail, this well established technique equips matrix indices with integer quantum numbers: On physical sites, these are the eigenvalues $q_n$ of local generators $g_n$ in a global symmetry of the form $\Gamma = \bigoplus_n g_n$. Virtual bond-indices may carry any accessible numbers $q_{n,n+1}$ selected by the MPS ansatz between sites $n$ and $n+1$. All non-zero matrix elements must further obey the additive fusion rule $q_{n-1,n} = q_n + q_{n,n+1}$; other combinations are removed from the ansatz. As a consequence, the MPS is confined to a fixed global sector $\Gamma |\Psi\rangle = q |\Psi\rangle$ of the symmetry, given by the difference $q := q_{L_x, L_x+1} - q_{0,1}$ of quantum numbers assigned to the first and last MPS bond.

We can now impose all constraints expressed by Eq. (5) in form of two global U(1) symmetries, which we generate locally from diagonal generators $g_n^{(k)}$ ($k \in \{0, 1\}$) taking the eigenvalues

$$q_n^{(k)} = \begin{cases} \mathrm{b}(\boldsymbol{r}_n) & \text{if } k \equiv n \pmod 2 \\ -\mathrm{b}(\boldsymbol{\ell}_n) & \text{else.} \end{cases} \tag{6}$$

The mapping $\mathrm{b}(\cdot)$ assigns a unique number to each configuration of spins, for instance by binary counting. The so defined $\Gamma^{(k)} = \bigoplus_n g^{(k)}$ generate all constraints between even-odd ($k = 0$) and odd-even ($k = 1$) pairs of adjacent sites. This is a consequence of the fusion rules at sites $n$ and $n+1$, which for $k \equiv n \pmod 2$ read

$$q_{n-1,n}^{(k)} - q_{n+1,n+2}^{(k)} = q_n^{(k)} + q_{n+1}^{(k)} = \mathrm{b}(\boldsymbol{r}_n) - \mathrm{b}(\boldsymbol{\ell}_{n+1}). \tag{7}$$

Eq. (5) is therefore equivalent to fixing quantum numbers on every other virtual bond to the same constant value, e.g. zero: $q_{n-1,n}^{(k)} = 0$ for $k \equiv n \pmod 2$ (see Fig. 1(c)). Once this

condition is enforced, the U(1)-invariant matrices of our MPS automatically attain a block-diagonal form [16], as we have now explicitly removed all redundant DOFs by discarding matrix elements that do not obey the fusion rule.

We have thus found a computationally efficient MPS representation that is constrained to a selected gauge sector. The only requirement is a standard MPS framework capable of handling global U(1) symmetries. With the $\Gamma^{(k)}$ being conserved quantities, such a framework will furthermore exactly preserve these constraints throughout an entire numerical simulation, such as a time-evolution or ground state search.

It is possible to incorporate further global Abelian symmetries into our ansatz (given they commute with each other). In particular, we keep the winding number $w_y$ fixed to the sector of the ground state in form of an additional U(1) integer quantum number $z_n = 2s_{(n,1/2)}$. Note that $w_x$ is already fixed by the open boundary condition $\ell_1$ and any static charges on the lattice. This allows us to further remove from our local basis $|i_n\rangle$ all configurations with incompatible net horizontal flux of arrows, and thus further mitigate the (still exponential) growth of the local MPS dimension $d_n = \dim\{|i_n\rangle\}$ with $L_y$. A brief account of matrix sizes, quantum numbers and their impact on computational cost is given in Appendix D.

Other invariances of the cylindrical lattice, within specific boundary conditions, include combinations of charge conjugation, inversion ($n \to -n$, $m \to -m$) and translation ($m \to m+1$). While possible in principle with a non-Abelian invariant MPS ansatz, we do not explicitly encode those symmetries in our simulations. More details on the choice of boundary conditions and topological sectors are provided in Appendix B.

## 2.4 Ground State Simulation

When mapping the 2D lattice problem onto a chain, the Hamiltonian Eq. (1) has only nearest-neighbour interactions, which are perfectly suited for many well established methods of MPS ground state search, such as DMRG [2], time evolution with the time-evolving block decimation (TEBD) algorithm [31,32], or time-dependent DMRG [49,50]. While all of these methods are ultimately limited by the sizes of bond- and local dimensions, and are sensitive to the energy spectrum in some way, they differ in how local updates of MPS matrices are carried out. In this paper we use imaginary time evolution via TEBD.

The TEBD algorithm requires us to provide nearest-neighbour evolution exponentials. To this end we reshuffle the interaction terms of Eq. (1) into sums over plaquettes at fixed $\hat{x}$-coordinate, which results in an open chain Hamiltonian $H = \sum_{n=1}^{L_x-1} H_{n,n+1}$. As is detailed in Appendix C, we then truncate the exponentials $e^{-\tau H_{n,n+1}}$ at first order in the time step $\tau$. The so obtained evolution operators $\tilde{U}_{n,n+1}(\tau) := \mathbb{I} - \tau H_{n,n+1}$ admit an efficient representation in terms of matrix products, keeping matrix sizes and computational cost manageable despite the large local basis.

We approximate ground states by means of imaginary TEBD as follows: First, we pick a maximal bond-dimension $\chi$ for the MPS and evolve an initial state with an initial time step $\tau_0$ as long as the energy difference between subsequent time steps exceeds a preselected threshold $\Delta E$. We then continue to evolve from that state at energy $E(\tau_0)$ with subsequently reduced step sizes $\tau_i = \tau_0 r^i$ (we found $r \approx 0.7$ to be a practical choice), until the differences between the so obtained final energies $E(\tau_i)$ drop below $\Delta E$ as well.

We have simulated $\mathcal{N}_s = (2L_x + 1)L_y \leq 590$ spins, in systems of size $L_x \times L_y$ up to $L_x = 30$ and $L_y = 10$. Our results have been tested for convergence in $\chi \to \infty$ and $\Delta E \to 0$ by subsequently improving both the bond-dimension up to $\chi = 2000$ and the convergence threshold down to $\Delta E = 10^{-12}$, with final time steps as small as $\tau \approx 2.3 \times 10^{-5}$. Exemplary results from this process are shown in Appendix E.

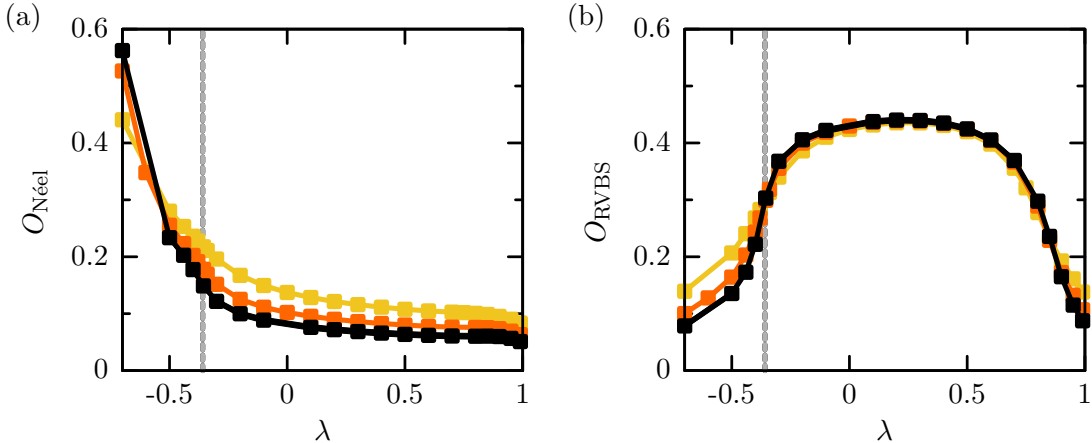

Figure 4: Order parameters (a) $O_{\text{Néel}}$ for the Néel phase and (b) $O_{\text{RVBS}}$ for the RVBS phase, plotted over the coupling parameter $\lambda$ for system sizes $L_x \times L_y = 18 \times 6$ (yellow), $24 \times 8$ (orange) and $30 \times 10$ (black). The vertical line indicates the location of $\lambda_c$ from [29].

# 3 Results: Order Parameters, Correlation Functions, and Wilson Loops

In this section, we discuss the phase diagram of the model, utilizing diagnostics based on order parameters, correlation functions, and Wilson loops. This analysis serves as a benchmark for our approach, by comparing known results to our extrapolation of the transition points, and by evaluating approximate string tensions from the decay of Wilson loops in real space. Furthermore, by investigating different aspect ratios $L_x/L_y$ as shown below, this comparison allows us to identify the best scaling regime to recover correct results by means of finite-size scaling.

While our algorithm is amenable to both periodic- and open boundary conditions along the $\hat{y}$-direction at a comparable computational cost, we opted for the cylindrical conditions in all simulations as to minimize finite-size and boundary effects.

## 3.1 Phase Transition from Néel to Resonating Valence Bond Solid

We first measured the ground state spin correlations, and evaluated the Néel-order parameter

$$O_{\text{Néel}} = \left\{ \mathcal{N}_1^{-1} \sum_{r \neq s} (-1)^{(r-s)\cdot(\hat{x}-\hat{y})} \langle \sigma_r^z \sigma_s^z \rangle \right\}^{1/2}, \tag{8}$$

where $r, s$ go over spin coordinates and $\mathcal{N}_1 = \mathcal{N}_s^2 - \mathcal{N}_s$ counts the number of expectation values summed up. Similarly, we detect the RVBS phase from staggered correlations between pairs of adjacent spins:

$$O_{\text{RVBS}} = \left\{ \mathcal{N}_2^{-1} \sum_{\nu,\mu} \sum_{\hat{i},\hat{j}} \sum_{\hat{k},\hat{l}} (-1)^{(\nu-\mu)\cdot(\hat{x}+\hat{y})} \langle \sigma_{\nu+\hat{i}/2}^z \sigma_{\nu+\hat{k}/2}^z \sigma_{\mu+\hat{j}/2}^z \sigma_{\mu+\hat{l}/2}^z \rangle \right\}^{1/2}, \tag{9}$$

where $\nu, \mu$ are lattice vertices, and $\hat{i}, \hat{j} \in \{\pm\hat{x}\}$, $\hat{k}, \hat{l} \in \{\pm\hat{y}\}$ are unit vectors. The sum goes over all $\mathcal{N}_2 = (4\mathcal{N})_\nu^2 - 4\mathcal{N}_\nu$ contributions, provided by four diagonal pairs around each of in total $\mathcal{N}_\nu = L_x L_y$ vertices, under the restriction that $\nu \neq \mu$ or $\hat{i} \neq \hat{j}$ or $\hat{k} \neq \hat{l}$. In order to reduce the

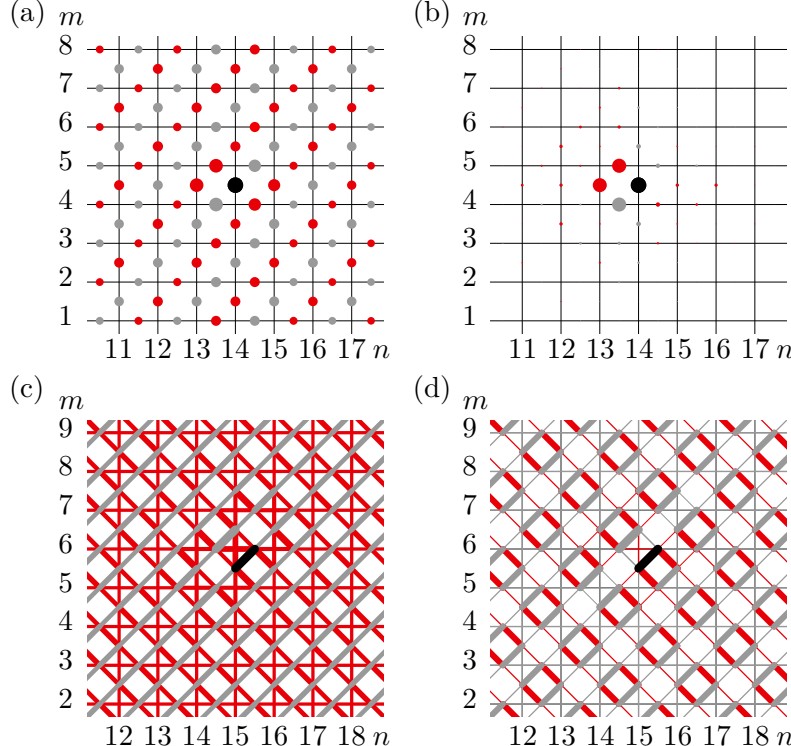

Figure 5: Spin-correlations visualized for a section of a $30 \times 10$ lattice in Néel- ($\lambda = -0.6$, left panels [(a),(c)]) and RVBS phase ($\lambda = 0.7$, right panels [(b),(d)]). Top panels [(a),(b)] show spin-spin correlations $\langle \sigma_r^z \sigma_s^z \rangle$ that enter $O_{\text{Néel}}$ in Eq. (8), while bottom panels [(c),(d)] show spin pair-pair correlations in $O_{\text{RVBS}}$ Eq. (9). Red and grey dots (lines for pairs) represent negative and positive expectation values of the correlation to the reference (black), their filled area (line-width for pairs) is proportional to the correlation strength (reference $= 1.0$).

impact of the open boundary, we further restricted the analysis to spin between sites $n_L = b$ and $n_R = L_x - b + 1$ with $b = 1$ or 2.

With these parameters, we can clearly distinguish both phases as shown in Fig. 4. The underlying two-point correlation functions are shown in Fig. 5, and already provide a qualitative picture of the different phases: In the top two panels, one can distinguish the very different behaviour of the $\langle \sigma_r^z \sigma_s^z \rangle$ correlation function, which is long-range ordered in the Néel phase, and short-ranged in the RVBS. The latter is instead characterized by the correlations between spin pairs, which indicate flippable plaquettes on a sublattice, as evidenced by Fig. 5(d).

Order parameters for the broken translational symmetry in the RVBS phase can also be derived from staggered flippability [28, 51]

$$O_{\text{flipp}} = \mathcal{N}_3^{-1} \sum_{\square} (-1)^{n+m} \langle f_{\square}^2 \rangle, \tag{10}$$

$$O_{\text{flipp2nd}} = \left\{ \mathcal{N}_4^{-1} \sum_{\square,\square'} (-1)^{n-n'+m-m'} \langle f_{\square}^2 f_{\square'}^2 \rangle \right\}^{1/2}, \tag{11}$$

where the summations go over all plaquettes $\square$ (and $\square'$) with respective lattice coordinates $n, m$ (and $n', m'$), normalized by $\mathcal{N}_3 = \mathcal{N}_{\square}$ and $\mathcal{N}_4 = \mathcal{N}_{\square}^2 - \mathcal{N}_{\square}$ for $\mathcal{N}_{\square} = (L_x - 1)L_y$ plaquettes. While the first order expectation works in our case as order parameter due to explicit symmetry breaking by boundary conditions, the second order correlations provide a more pronounced result.

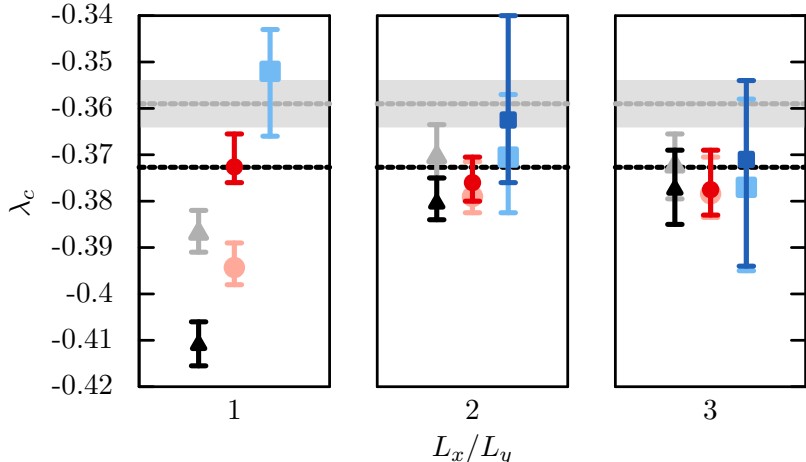

Figure 6: Finite-size scaling results for the transition point $\lambda_c$, from Néel and RVBS order parameters (red circles: $O_{\text{RVBS}}$, blue squares: $O_{\text{Néel}}$; dark red/blue: spin within distance $b = 2$ to the open boundary have been excluded) and flippability (grey triangles: $O_{\text{flipp}}$, black triangles: $O_{\text{flipp2nd}}$). Error bars are due to simulation precisions (i.e. finite bond-dimension). System sizes have been scaled up to widths $L_y \in \{6, 8, 10\}$, held at three fixed aspect ratios $L_x/L_y \in \{1, 2, 3\}$. Dashed lines indicate the results from [29] (grey with uncertainty) and [28] (black).

From finite size scaling [52, 53] over lattices of widths $L_y \in \{6, 8, 10\}$, kept at constant aspect ratios $L_x/L_y \in \{1, 2, 3\}$, we locate the transition point at $\lambda_c = -0.37(3)$ (see Fig. 6). This is consistent with previous exact diagonalization results $\lambda_c = -0.359(5)$ [29] and $\lambda_c \simeq -0.3727$ [28].

In detail, our values reported in Fig. 6 have been found from the intersection of curves $O(\lambda; L_y) \times L_y^{\gamma}$, where $O$ is one of our order parameters, rescaled for the different widths $L_y$. To this end, the exponent $\gamma$ was tuned to make all three curves intersect at the closest possible values $\lambda$. The reported error-bars account for the uncertainty in determining the intersection, which is also limited by simulation errors affecting the value of $O$ (see Appendix E for estimates), which around $\lambda_c$ tend to be dominated by limitations in the bond-dimension.

From the differences between our results at the two smallest ratios, where systems are limited to lengths $L_x = 10$ and $L_x = 20$ respectively, we can estimate our uncertainty due to the limiting maximal width $L_y = 10$. Compared to the length, further increasing the width should however have a much smaller impact due to the periodic boundary conditions. We note that the estimates at ratio $L_x/L_y = 1$ display severe finite size effects, while geometries with $L_x/L_y = 2$ and 3 provide much better agreement between the different order parameters, as they optimally combine large system sizes with modest anisotropies.

## 3.2 Confining Properties: Wilson Loops and String Tension

We now turn to a diagnostic which is directly related to confinement — i.e., Wilson loops [54]. In typical lattice gauge theory formulations in Euclidean space-time, Wilson loops are defined as the product of parallel transporters over a spatio-temporal closed path. Differently from traditional Monte Carlo simulations of lattice actions, wave-function based methods such as GITN can only access real-space Wilson loops. As such, extracting the string tension from the latter comes with a caveat, as these two types of Wilson loops coincide only in the presence of Lorentz invariance. We comment on this aspect at the end of the next section.

In our simulations, we measured the expectation values of the operator that flips oriented

loops of spins on the boundary of a rectangular area $A$ with $n \in [n_1, n_2]$, $m \in [m_1, m_2]$,

$$f_A = \prod_{n=n_1}^{n_2-1} \prod_{m=m_1}^{m_2-1} \sigma^+_{(n_1,m+1/2)} \sigma^+_{(n+1/2,m_2)} \sigma^-_{(n_2,m+1/2)} \sigma^-_{(n+1/2,m_1)} + \text{H.c.} \tag{12}$$

The connection to the more traditional definition of the Wilson loop is immediately established considering that the present model is a quantum link formulation of a compact U(1) gauge theory [29,45]. For a single plaquette $A = \square$, one recovers the plaquette operator $f_\square$.

In order to smoothen out corner contributions (which will lead to additional corrections due to ultra-violet effects), we tried to employ Creutz's ratios [55]. However, we found out that the latter did not facilitate our analysis, probably due to the fact that we could only access modest sizes of the area enclosed by the Wilson loop itself.

Instead, in systems of dimension $L_x = 3L_y$, with $L_y \leq 10$, we fit the expectations $W(A) = \langle \Psi_{\text{GS}} | f_A | \Psi_{\text{GS}} \rangle$ to an exponential decay with the enclosed area

$$W_{\text{fit}}(A) = \exp\left\{-\alpha_{\Delta_y} A + \beta_{\Delta_y}\right\}, \tag{13}$$

where $A = \Delta_x \Delta_y$ is of width $\Delta_x = n_2 - n_1$ and height $\Delta_y = m_2 - m_1$, and both the area contribution $\alpha_{\Delta_y}$ as well as $\beta_{\Delta_y}$ are free parameters that may depend on $L_y$ and $\Delta_y$ (besides $\lambda$) in order to account for finite size corrections and the sharpness of the boundaries. Typical results for two points ($\lambda = -0.4$ and $0.6$) are shown in Fig. 7(a).

Overall, we find that the numerical results are very well captured by Eq. (13) even for modest areas. For the accessible system sizes, the dependence of $\alpha_{\Delta_y}$ and $\beta_{\Delta_y}/\Delta_y$ on $\Delta_y$ appears to be of the type $c_0 + c_1 \text{atan}(c_2/\Delta_y)$ with some coefficients $c_k$. Remarkably, the exponential decay of the Wilson loop seems to be reproduced extremely well even when the value of the correlation itself is smaller than our truncation error. This indicates that, given a certain truncation error, the corresponding error in the Wilson loop correlation can be considerably smaller. Nevertheless, the slopes $\alpha_{\Delta_y}$ remain affected by simulation precision and we extrapolated our results in both $\chi$ and $\Delta E$ (see Appendix E). The reason is that even modest variations of the Wilson loop correlator affect the estimate of $\alpha_{\Delta_y}$.

In order to characterize the area contribution $\alpha_\infty$ for large $\Delta_y$ and in the thermodynamic limit (TL) $L_y \to \infty$, we fixed the rectangle width at half the system size, $\Delta_y = L_y/2$, and plotted $\alpha_{L_y/2}$ over $\lambda \in [-0.6, 1]$ for $L_y = 4, 6, 8$ and $10$ (see Fig. 7(b)). Apparently, the value of $\alpha_{L_y/2}$ generally decreases with $L_y$ — most prominently around $\lambda_c$ and $\lambda_{\text{RK}}$ due to finite size effects, while the decline occurs at a slower rate in the bulk of the RVBS where we expect a finite $\alpha_\infty > 0$. Even though we cannot conclusively determine the asymptotic behaviour from the available system sizes, we estimated $\alpha_\infty$ from linear extrapolation to $1/L_y$ (black crosses in Fig. 7(b)). As expected, the value of the area contribution approaches zero when moving towards the deconfining RK-point $\lambda_{\text{RK}} = 1$. Moreover, we observe a visible drop of the decay coefficient for $L_y > 6$ in the almost-critical region around $\lambda_c$, while the TL estimates suggest a finite value well above zero — testifying that there is a finite mass gap at the transition point.

## 3.3 String Tension from the Potential Energy between Static Charges

A direct estimate of the string tension is provided by introducing two static charges in the system, and monitor the amount of excess energy, compared to the ground state in absence of charges, as a function of their distance from each other. In our simulations, we considered a lattice with two static charges $\bar{c}_+ = +1$ and $\bar{c}_- = -1$, placed at the sites $n_\pm = (L_x + 1)/2 \pm \ell/2$ in horizontal distance of $\ell = n_+ - n_-$ lattice spacings. This configuration has been chosen in order to alleviate potential boundary effects (both charges are at the same distance from the boundaries). At the MPS level, these static charges can be readily implemented by changing the value of the Gauss-law at a given site.

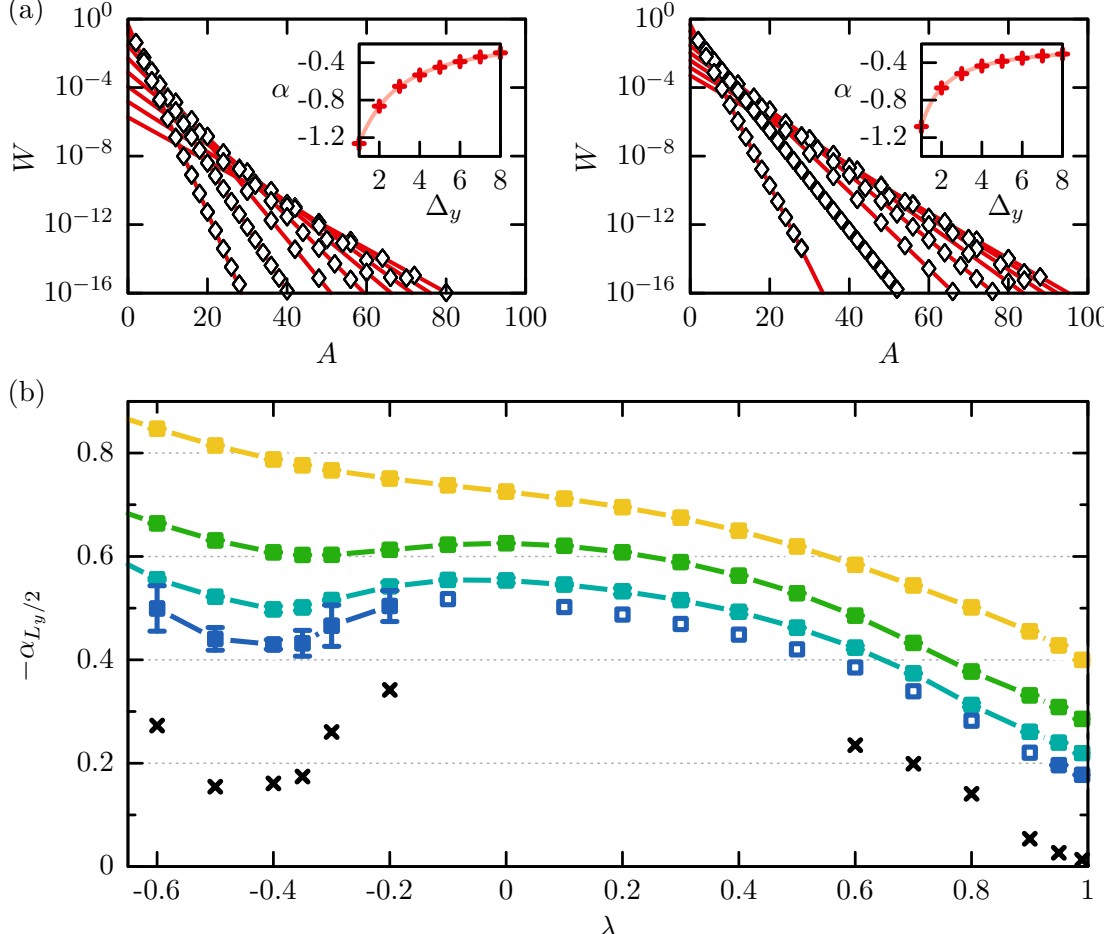

Figure 7: (a) Wilson loop expectations $W$ over enclosed area $A = \Delta_x \Delta_y$ for $\lambda = -0.4$ (left) and $\lambda = 0.6$ (right), in a $30 \times 10$ lattice simulated at $\chi = 1500$. Different lines correspond to different values of $\Delta_y$. Slopes $\alpha_{\Delta_y}$ from linear regressions to $\log W$ at fixed $\Delta_y$'s (red lines) are fitted by an atan-logistic (inset, rose) (see text). (b) Slopes $\alpha_{L_y/2}$, extrapolated in $\chi \to \infty$ (except open squares: $\chi = 1500$, error $\propto$ symbol size) over $\lambda$, from lattices $L_x = 3L_y$, $L_y = 4, 6, 8, 10$ (yellow, green, cyan, blue). Estimates $\alpha_\infty$ for $L_y \to \infty$ are indicated without errors (black crosses) as system sizes are too small to conclusively determine the asymptotic behaviour.

In Fig. 8(a), we show a typical result for the local energy distribution in the cylinder along the $\hat{x}$-axis. The plot already shows how the insertion of static charges significantly affects the expectation value of the Hamiltonian density, not only close to the charges themselves, but in the entire region between the two.

The string tension is evaluated by considering the energy difference between the system ground states with ($E^c$) and without ($E^0$) static charges, and by plotting it as a function of the intercharge distance $\ell$:

$$E^c - E^0 = \ell\sigma + ... \quad . \tag{14}$$

Two typical sets of results are presented in Fig. 8(b): Already for modest values of $\ell$, the linear dependency captures the numerical data well.

The fitted values of $\sigma$ are shown in Fig. 8(c) for the parameter regime $\lambda > 0.2$. As expected, the string tension vanishes at the RK point, and increases almost linearly within the RVBS phase. However for $\lambda \ll 1$, finite size effects, that confine the string and lead to self-

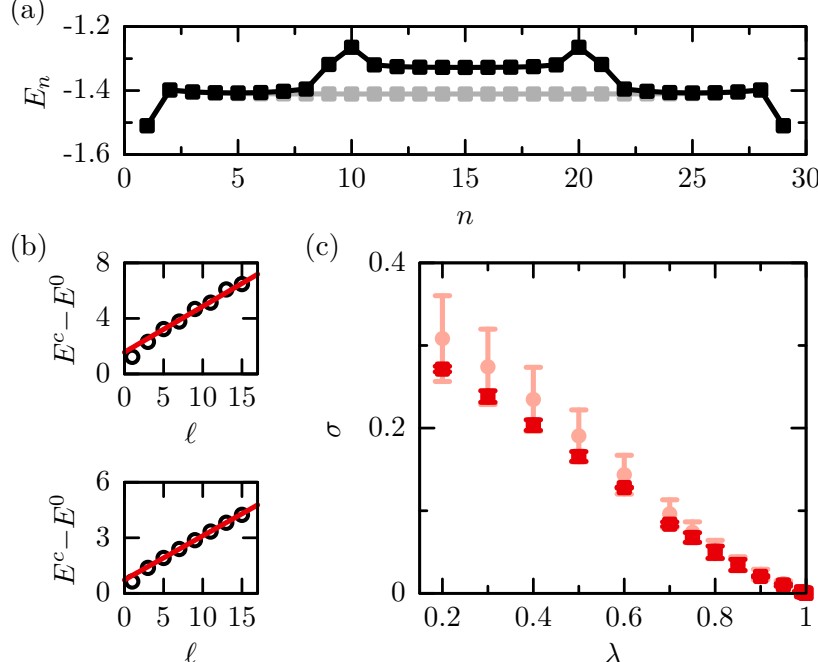

Figure 8: String tension from potential energy between two static charges $\pm 1$. (a) Cut through the energy landscape at $\lambda = 0.7$, size $30 \times 10$ from plaquette energies $E_n = \sum_m E_{m,n}$ with charges at $n_- = 10$, $n_+ = 21$ ($\ell = 11$) (black) and without charges (grey). (b) Excess energy over intercharge distance $\ell$, fitted linearly (red) for $\lambda = -0.3$ (top) and $\lambda = 0.3$ (bottom) in the $24 \times 8$ lattice. (c) Fit-slopes $\sigma$ in the RVBS phase, for sizes $24 \times 8$ (red) and $30 \times 10$ (rose).

interaction, become visible with nonlinearities in Eq. (14) and values $\sigma$ start to drift with increasing system size. Eventually our system sizes become too small for a direct measurement of the string tension for $\lambda \lesssim 0.2$, including $\lambda_c$. We note that the string tension results are in reasonably good qualitative agreement with the values of $\alpha$ extracted from the expectation values of the Wilson loops in the previous section. This indicates that, despite the RVBS order breaks translational symmetry at the lattice spacing level, the breaking of Lorentz invariance affects the expectation value of the Wilson loop, but not its parametric dependence on $\lambda$. At the quantitative level, it is instead impossible to draw conclusions on this parallelism: The reason is that, as discussed in the last subsection, the system sizes we simulated are too small to properly extrapolate $\alpha_{L_y}$ to the thermodynamic limit.

## 4 Entanglement Properties of the Ground State and String Excitations in the Resonating Valence Bond Solid Phase

Tensor network simulations provide direct access to the entanglement properties of the ground state and of the low-lying states of the theory. In the context of the quantum ice model, these properties are essentially unexplored, yet intriguing: Given the fact that this model describes a gauge theory, its Hilbert space cannot be written as a tensor product for bipartitions of any size while upholding the Gauss-law.

In this section, we first discuss the entanglement spectrum (ES) properties of the RVBS phase, and then turn to the entanglement entropies of string excitations.

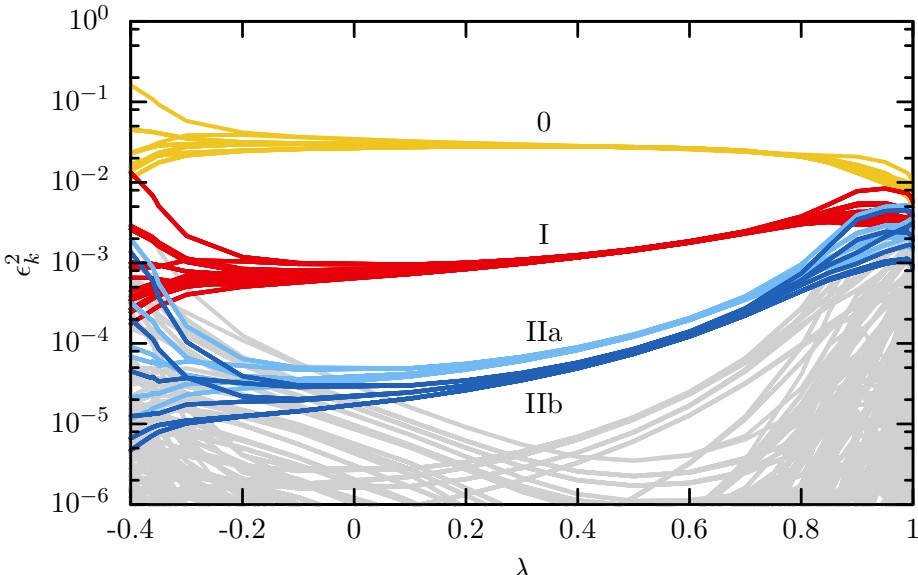

Figure 9: Entanglement spectrum of the RVBS: Reduced density matrix eigenvalues from a vertical cut through a $30 \times 10$ cylinder between sites $n = 10, 11$ are plotted over the coupling $\lambda$. In the bulk of the RVBS phase, degenerate levels 0, I, IIa and IIb (yellow, red, bright and dark blue) are highlighted.

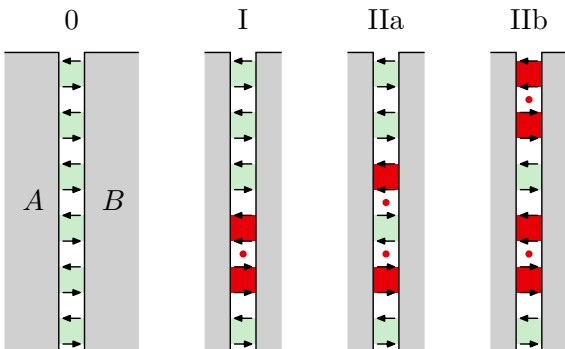

Figure 10: Spin configurations $s_{(m,n+1/2)}$, for a vertical cut $m = 1, \ldots, L_y$, that we associate with the lowest levels 0–IIb of the entanglement spectrum in the RVBS (see Fig. 9). Oriented plaquettes on the resonant sublattice (green) can be destroyed (red) by virtual plaquette-flips in the opposite sublattice (red dots).

## 4.1 Entanglement Spectrum in the Resonating Valence Bond Solid Phase

We characterize the entanglement properties of the system by investigating the reduced density matrix of a bipartition $A$:

$$\rho_A = \text{Tr}_B \, |\text{GS}\rangle\langle\text{GS}| = \sum_k \epsilon_k^2 |\phi_k\rangle\langle\phi_k|, \tag{15}$$

obtained by considering the ground state wave function, and taking the partial trace with respect to the complement of $A$, which we denote as $B$.

In our MPS representation, we have direct access to the ES in form of the singular values $\{\epsilon_k\}$ associated with a cut through the cylinder between any sites $n$ and $n+1$. A sample of our results from the largest system size we simulated is shown in Fig. 9, where the cut is made at an off-center MPS bond $n < L_x/2$ in order to reveal degeneracies in the spectrum that are not

merely caused by lattice(-inversion) symmetries. We can characterize each level by the spin configuration on the horizontal links joining the bipartition. According to this criterion, we have grouped states 0-IIb (see Fig. 10):

Level 0 comprises all $d_0 = 2^{L_y/2}$ configurations available under the condition that spins are anti-aligned on the sublattice hosting the resonant, oriented plaquettes. A subsequent flip of a single pair of antiparallel spin on any of the $L_y/2$ plaquettes of the opposite sublattice leads to the boundary configurations of level I, which is $d_I = 2^{L_y/2-2}L_y$ degenerate. This subspace describes 'excitations' on top of the RVBS state.

In levels IIa and IIb, two such defects occur, reducing the number of antiparallel alignments on the resonant sublattice to $L_y/2-2$ or $L_y/2-4$, depending on whether the defects are adjacent or not. Corresponding degeneracies are $d_{IIa} = 2^{L_y/2-3}L_y$ and $d_{IIb} = 2^{L_y/2-4}L_y(L_y/2-3)$, respectively. The drop in singular values with each level is roughly proportional to the number of defects.

This spectrum gives a direct signature of the RVBS state. In particular, all counting above just reflects the structure of the symmetry broken state, and excitations on top of that. In principle, one can try to apply finite-size scaling analysis to extract the transition point $\lambda_c$ from the ES: However, our results suggest that the latter suffers from more severe finite size effects than the correlation functions, rendering this procedure rather inaccurate.

## 4.2 Entanglement Entropy of String Excitations

String excitations on top of the vacuum play a paradigmatic importance in lattice field theory. Historically, they have been vastly employed to provide direct information about confining properties of a theory (without the need of adding dynamical charges) [56]. In the context of magnetism, string excitations have been widely discussed as a distinctive feature of classical spin ices [57, 58], and some of their properties have been recently observed [59, 60] and discussed at the quantum level [61, 62].

While string excitation are typically addressed employing diagnostics based on energetics (such as, for instance, the analysis of Lüscher's terms in the string potential), we employ here a direct investigation of the string entanglement properties to show how, in the RVBS phase, string excitations are well captured by a conformal field theory with central charge $\bar{c} = 1$.

In order to estimate the central charge, we have evaluated the entanglement properties in form of the bipartite von Neumann entropies $S_{\pm 1}(n, \ell)$ in presence of charges $\pm 1$ in distance $\ell$, from the MPS bond between sites $n, n + 1$ (same configuration as the one employed to investigate the string tension). From these results, we have subtracted the background values from simulations without charges $S_{BG}(n)$. The difference $S_{diff}(n) = S_{\pm 1}(n, \ell) - S_{BG}(n)$ is then interpreted as the string entanglement entropy.

Finally, we fit our numerical results with [63]

$$S_{fit}(n) = \frac{c}{6} \log\left[\frac{\ell - 1}{\pi} \sin\left(\pi \frac{n - n_- - 1}{\ell - 1}\right)\right] + S_0 \,, \tag{16}$$

where the free fit parameters are $c$ (central charge) and offset $S_0$. Two typical sets of result are shown in Fig. 11(a). Despite the small number of points available even for our largest system size, we observed an accurate fit of our data (in one-dimensional systems, Eq. (16) is sometimes not accurate at distances of order of the lattice spacing due to strong UV corrections).

When approaching the transition point $\lambda_c$ with $\lambda \le -0.2$, these results tend to become unreliable and deviate with increasing system size, as the string excitations start spreading over a region of space in the $\hat{y}$-direction which is of the order of $L_y$. Instead, deep in the RVBS phase, we observe that, for our largest system size, the string width is sufficiently small such that self-interaction effects are negligible. In this regime, the value of the central charge (blue

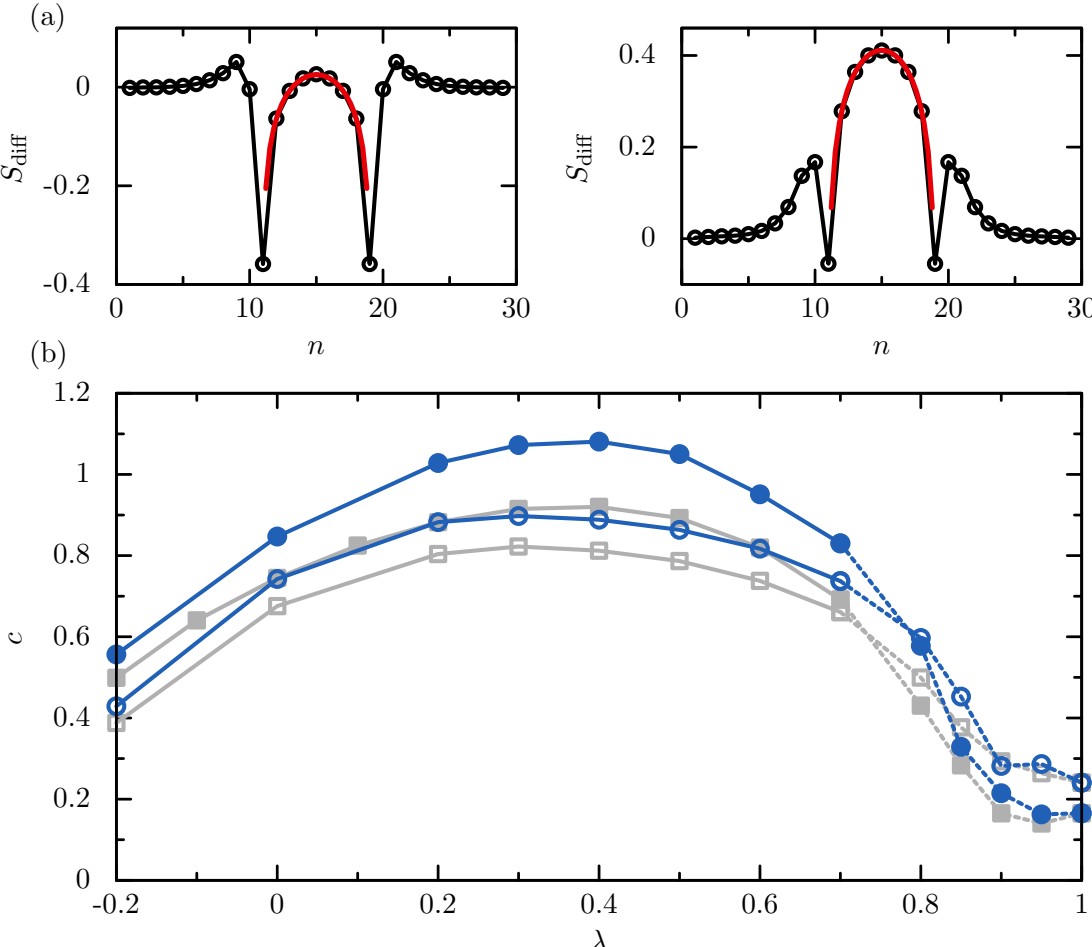

Figure 11: (a) Increase in von Neumann entropy after insertion of two static charges $\pm 1$ in distance $\ell = 9$ on a $30 \times 10$ lattice. Entropies taken at vertical cuts of the cylinder between sites $n, n+1$. Between the charges, a Calabrese–Cardy fit (red) yields a central charge $c$ as shown for $\lambda = -0.2$ (left) and $\lambda = 0.7$ (right). (b) Central charges $c$ over coupling parameter $\lambda$ for system sizes $24 \times 8$ (grey boxes) and $30 \times 10$ (blue points), $\ell = 5$ (open) and $\ell = 9$ (filled symbols). For $\lambda \in [0.7, 1)$ (dashed lines), resonant plaquettes on both sublattices where found in superposition when charges where present, increasing the entanglement entropy compared to the system without charges. For $\lambda = 1$, exact ground state results are shown. Bare fit-errors are smaller than point sizes.

dots in Fig. 11) is close to 1 within the fit error (which is of order 10%, including the bare fit error and an estimate due to the few points available). This observation is consistent with a conformal field theory description of the string excitations in terms of a compactified boson theory. While this behaviour, in the context of non-Abelian gauge theories, has already been widely discussed [33, 64], this was so far only diagnosed based on long-distance behaviour of particle-antiparticle potentials. Here, we have instead provided an entanglement-based proof of this behaviour, that has the key advantage of requiring more modest system sizes to be precisely diagnosed.

# 5 Conclusion

We have reported a gauge invariant tensor network investigation of square ice — a U(1) quantum link model in two spatial dimensions. Our results on the phase diagram are quantitatively consistent with previous results in the literature, and represent an important systematic benchmark for the accuracy and reliability of tensor network methods applied to confining lattice gauge theories in more than one dimension.

In addition to diagnostics based on correlation functions and the excitation spectrum, we have investigated the entanglement properties of the system. First, we have studied the entanglement spectrum of the ground state wave function: While being too finite-size sensitive to be employed for estimating phase transition points, it reflects the characteristic symmetry breaking pattern of the RVBS phase. Secondly, we have provided evidence that the string excitations in such models behave as bosonic strings, using diagnostics based on the entanglement entropy. This prediction can be in principle tested in quantum simulators for quenched lattice gauge theories [51, 65–67], and testifies the fact that universal properties of gauge theories (such as the conformal description of string excitations) can be observed in such systems despite achieving modest system sizes. From the theoretical viewpoint, our results motivate the search for a microscopic derivation of the bosonic string theory of U(1) quantum link models that could complement the numerical results presented here, and possibly extend those to parameter regimes in the vicinity of the transition point between RVBS and Néel phases.

Our work indicates that, whilst challenging from a computational physics perspective, tensor network methods for the simulation of confining lattice gauge theories can reach system sizes which allow a distinct, non-ambiguous characterization of phases of matter. While our study here was on a quenched gauge theory for which accurate Monte Carlo simulations can be carried out — so that we could have a direct benchmark —, the tensor network methods we employ are sign-problem free, and can be applied to real-time dynamics and to theories in the presence of fermionic matter. As a direct extension of this work, it would be interesting to study the effect of dynamical matter in U(1) quantum link models, which is also connected to recently formulated problems in the context of strongly correlated electrons [68, 69]. From the methodological perspective, testing the performance with more specialized and powerful tensor network classes such as projected entangled pair states [6, 19–22, 70, 71] or two-dimensional tree tensor networks [4, 72–74] appears promising in view of the entanglement results we presented.

# Acknowledgements

We thank D. Banerjee, C. Castelnovo, K. Penc, E. Rico, N. Shannon, P. Silvi, K. Van Acoleyen, and U.-J. Wiese for discussions.

**Funding information**   FT gratefully acknowledges support from the Carl-Zeiss-Stiftung via Nachwuchsförderprogramm, and the state of Baden-Württemberg through bwHPC. This work is partly supported by the Baden-Württemberg Stiftung via Eliteprogramm for PostDocs, the EU-Flagship project PASQUANS, the QUANTERA project QTFLAG and the German Research Foundation (DFG) via a Heisenberg fellowship and TWITTER. MD work is partly supported by the ERC Starting grant AGEnTh (758329).

# A  Efficient Gauge Invariant Tensor Networks for Quantum Link Models

This section explores the efficient encoding of GITNs by means of U(1) invariant tensors in more general terms. As a special case, we obtain the MPS of Sec. 2.3.

We first make the connection to the underlying construction of GITNs, as put forward in Ref. [5]: It is based on the QLM formulation [34,75,76] of Abelian and non-Abelian Hamiltonian LGTs on arbitrary lattices in $D$ dimensions, however we limit ourselves here to the case of rectangular lattices with unit lattice vectors $\hat{u}$ along dimensions $u \in \{1,\dots,D\}$. Gauge-symmetry generators $\{G_\nu\}$ then commute with the Hamiltonian $H$ at every vertex $\nu$, and bosonic gauge fields $U_\mu^{a,b}$ live on the links $\mu = \langle \nu, \nu + \hat{u} \rangle$ between adjacent vertices. The generalized 'color'-indices $a, b$ only appear in the presence of non-Abelian gauge symmetries. In the QLM formalism, on each link, the field operators are substituted with two 'rishons', one of which we attribute to either end of the link: $U_\mu^{a,b} = c_{\nu+\hat{u},-\hat{u}}^{a\dagger} c_{\nu,+\hat{u}}^{b}$. This mapping introduces an artificial local U(1) symmetry, in that the number of rishons per link $N_\mu = n_{\nu+\hat{u},-\hat{u}} + n_{\nu,+\hat{u}}$ with $n_{\nu,\pm\hat{u}} = \sum_a c_{\nu,\pm\hat{u}}^{a\dagger} c_{\nu,\pm\hat{u}}^{a}$, is conserved and can be fixed by choosing a specific $\bar{N}_\mu$-dimensional representation for the gauge bosons. In numerical simulations, the $\bar{N}_\mu$ are necessarily finite, and give rise to the *link-constraints* $N_\mu |\psi\rangle = \bar{N}_\mu |\psi\rangle$. Additionally, any GITN state $|\psi\rangle$ must obey the *gauge-constraints* $G_\nu |\psi\rangle = 0$ (which extend to gauge-covariance by fixing nontrivial representations on the right hand side, such as the U(1)-charges in Eq. (3)). Since the generators $\{G_\nu\}$ only act on matter- and rishon modes $c_{\nu,\pm\hat{u}}^{a}$ at site $\nu$ and are furthermore diagonal in the rishon-number of each link, one can select a gauge-invariant local basis $|i_\nu\rangle$ fulfilling $G_\nu |i_\nu\rangle = 0$, and write $|i_\nu\rangle = \bigotimes_u |n_{\nu,-\hat{u}}\rangle |n_{\nu,+\hat{u}}\rangle |\varphi_\nu\rangle$ in terms of the rishon-numbers associated with site $\nu$. With remaining degeneracy and matter modes captured by $\varphi_\nu$, these local states constitute a canonical computational basis for GITNs. The link-constraints however remain to be enforced by other means.

The idea pursued in this paper, and previously for a non-Abelian LGT in 1D [12], employs abelian symmetry invariant tensor networks for a numerically exact and efficient encoding of link-constraints in form of a selection rule on shared quantum numbers. Let's assume we work with a MPS [44] in 1D, or more generally a PEPS [70,71,77,78] in $D$ dimensions, such that we have a tensor at each vertex $\nu$ equipped with one physical index $i_\nu$ in the computational basis, and $2 \times D$ virtual- or bond-indices. The latter indices are shared with adjacent tensors at the vertices $\nu \pm \hat{u}$, following the pattern of links on the lattice. Let us further separate all vertices into two sublattices, denoted by $k \in \{0,1\}$, such that vertices of either sublattice are no longer adjacent. We can then encode all the local U(1)-symmetries that conserve the rishon number at every link, by using a network ansatz with inbuilt *global* U(1) invariances [46,47] and a set of just $2 \times D$ quantum-numbers $\{q^{(k,u)}\}$, with $k$ being a sublattice- and $u$ a dimension index. All tensor indices, no matter whether they are physical for site $\nu$ or virtual for bond $\langle \nu, \nu \pm \hat{u} \rangle$, can then be written in terms of these quantum numbers (plus one additional degeneracy index), and tensor elements at site $\nu$ obey the fusion-rules $\sum_w q_{\langle \nu, \nu-\hat{w} \rangle}^{(k,u)} = q_\nu^{(k,u)} + \sum_w q_{\langle \nu, \nu+\hat{w} \rangle}^{(k,u)}$; otherwise they are zero and need not be stored. Specifically, we may choose to associate the indices $i_\nu$ of physical sites with quantum numbers

$$q_\nu^{(k,u)} = 2n_{\nu,\pm\hat{u}} - \bar{N}_{\langle \nu, \nu\pm\hat{u} \rangle} , \tag{17}$$

where in double-symbols, '+' shall apply if $\nu$ is part of the $k$'th sublattice, and '−' otherwise. The link-constraints can then be enforced by simply fixing all quantum numbers associated with the network's bond-indices to zero, with the exception of $q_{\langle \nu, \nu\pm\hat{u} \rangle}^{(k,u)}$. In doing so, the fusion-rules simplify to $q_{\langle \nu, \nu\pm\hat{u} \rangle}^{(k,u)} = \mp q_\nu^{(k,u)}$, and together with Eq. (17) those quantum numbers directly and controllably encode the state of the gauge-bosons in the network's bond-indices. Combining

the rules from both ends of a link $\mu = \langle v, v+\hat{u} \rangle$ reproduces $\bar{N}_\mu = n_{v,+\hat{u}} + n_{v+\hat{u},-\hat{u}}$. Furthermore, once the required set of quantum numbers is fixed to zero, these numbers, and in consequence all link-constraints, are dynamically protected due to $[N_{v,v+\hat{u}}, H] = 0$. It is therefore possible to precisely and permanently parameterize the gauge-invariant (or a covariant) sector of the Hilbertspace with ordinary TNs, such as MPS or PEPS, merely by formatting all input tensors in terms of the above defined quantum numbers. As long as the underlying numerical framework supports global abelian U(1) symmetry invariances, these GITNs can directly be used for efficient simulation with standard TN algorithms, including time-evolution or ground-state search.

For TNs that do not mirror the geometry or spatial dimension of the lattice, similar constructions can be made. It is important to lay out the specific network first, and then introduce the required global abelian symmetries to take care of all link-constraints. In particular, the gauge-invariant MPS of Sec. 2.3 is based on a projection of the spin ice from Eq. (1), an originally two-dimensional U(1) QLM with $\bar{N}_\mu \equiv 1$ [5], onto a one-dimensional chain. In this process, all vertical link-constraints become local selection rules in the computational basis, but the constraints from horizontal links remain to be enforced between neighbouring sites of the chain. By closely following the above construction, for each such pair of MPS sites, we must independently conserve the number of $L_y$ distinct rishon species (namely, the rishons $n_{v,+\hat{x}}$ and $n_{v+\hat{x},-\hat{x}}$ from all horizontal links in the original QLM with same $\hat{x}$-coordinates). To this end, these rishons can be encoded by $2L_y$ independent U(1) quantum-numbers, repeating the procedure of Eq. (17) for each species with $k \in \{0, 1\}$ and $D = 1$. However, one can always get away with fewer quantum numbers with the help of an isomorphism, that maps several integers from ranges $\{0, \ldots, \bar{N}_\mu\}$ onto scalars, and preserves the component-wise additive fusion-rule. In our case, the binary encoding in Eq. (6) maps all rishon species onto just two global quantum numbers $q^{(k)}$. In general, such a mapping might also come in handy when working in tensor network frameworks that only support a single global symmetry.

# B  Topological Sectors and Boundary Conditions

This section discusses the topological sectors and boundary conditions (BCs) exploited in our TEBD simulations, both of which are directly controlled by the encoded global U(1) quantum numbers. For instance, the spin on the boundary are fixed in $q_{0,1}^{(k)}$ and $q_{L_x, L_x+1}^{(k)}$ on the first and last MPS bond along with $w_x$. Similarly, we can fix the global winding number $w_y$ [27]. We now restrict ourselves to net total charge zero and even lattice dimensions $L_x$, $L_y$, and we set $s_{(1/2, m)} = (-1)^m/2$ at the left boundary as to recover the ground-state winding number $w_x = 0$. On the opposite end, we can choose *aligned* conditions $s_{(L_x+1/2, m)} = s_{(1/2, m)}$, as well as *anti-aligned* conditions $s_{(L_x+1/2, m)} = -s_{(1/2, m)}$ (see Fig. 12(a)). With either choice, we target the respective lowest energy state in topological sector $w_y = 0$. However, for ground-state simulations in the absence of charges, e.g. in the ice-rule manifold, we select the anti-aligned conditions as they are best suited when focusing on the RVBS phase. This is due to the existence of a finite-size gap [28, 29], related to the spontaneous breaking of translational symmetry, which complicates straightforward converge towards a well defined ground state via imaginary time evolution. As shown in Figs. 12[(b)-(d)] from exact diagonalization, these gaps depend on the boundary conditions. In the RVBS phase, the largest gap with slower decay in even cylinder length $L_x$ is the one corresponding to anti-aligned BCs (compatible with a slower power-law decay, however exponential behaviour cannot be ruled out from data).

These practical observation can be traced to explicitly broken lattice symmetries (not yet controlled via global quantum numbers) at the boundary: In general, the cylindrical square ice of Eq. (1) is invariant under combinations of charge-conjugation on horizontal- and vertical

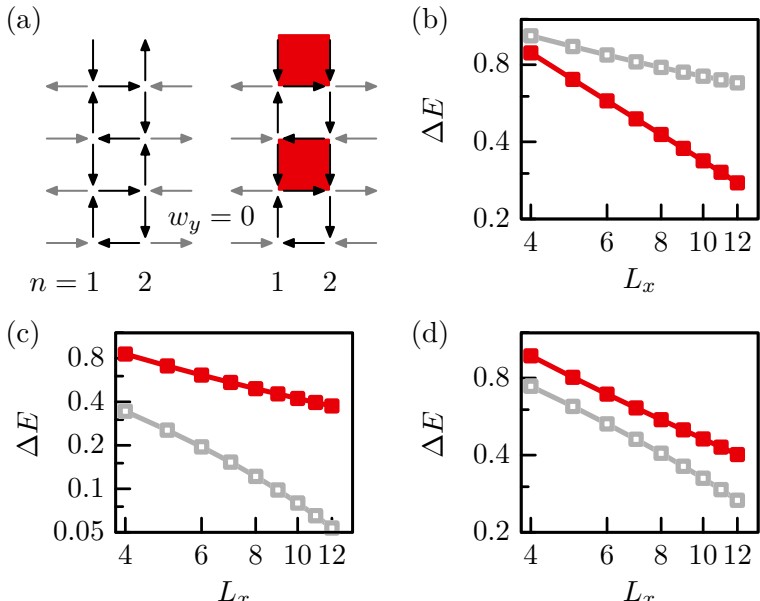

Figure 12: (a) Two choices of boundary conditions (grey arrows) illustrated on a $2 \times 4$ lattice (left: 'aligned', right: 'anti-aligned'). 'Anti-aligned' conditions prevent oriented plaquettes from fully covering one of the two sublattices (red) in the ground-state sector of the winding number, $w_y = 0$. [(b)-(d)] Energy gap (in log-log scale) between the two lowest energy states within the ground-state sector $w_y = 0$ ($|w_y| = 1/2$ for odd $L_x$), from exactly diagonalized $L_x \times 4$ grids. For even $L_x$, red filled boxes indicate anti-aligned BCs, while grey open boxes indicate aligned BCs. For odd $L_x$, colors are swapped. (b) $\lambda = -0.5$, (c) $\lambda = -0.3$, (d) $\lambda = 0.3$.

bonds (inverting spins $s_\mu \to -s_\mu$ on respective bonds), lattice-translation ($T_y : m \to m+1$) and lattice-inversions ($I_x : n \to L_x - n + 1$ and $I_y : m \to L_y - m + 1$). With 'aligned' BCs, permissible operations are for instance $P_1 = C_h \times I_y$, $P_2 = C_v \times I_x$ and $P_3 = C \times T_y$, while 'anti-aligned' conditions instead support $Q_1 = P_1$, $Q_2 = C \times P_2$ and $Q_3 = P_3^2$. The latter explicitly breaks the (otherwise spontaneously broken) translational symmetry that transfers resonant plaquettes of the RVBS from one sublattice onto the other: Namely, oriented (and resonating) plaquettes may fully cover only one of the two sublattices, while the other sublattice can not be oriented fully (with at least $\propto L_y$ defects at the boundary, as depicted in Fig. 12(a). As a consequence, resonances on the former sublattice are preferred without frustrating the resonant structure, and the finite size gap is increased. Conversely, the two symmetry breaking Néel configurations remain coupled by $Q_2$. Instead, in the Néel phase $\lambda \lesssim \lambda_c$ the gap can only be increased by selecting 'aligned' BCs, under which no such transition is permitted and in similar fashion, only one of the two conjugate Néel-configurations can maximize the number of oriented plaquettes.

## C  Imaginary Time Evolution with Time-Evolving Block Decimation

Here we describe the details of our TEBD implementation, especially the mapping of the Hamiltonian into nearest-neighbour form. We begin by rewriting the plaquette-operator into our computational basis $|i_n\rangle$ as introduced in Sec. 2.3. A plaquette spanned by lattice-coordinates $\{n, n+1\} \times \{m, m+1\}$ can be flipped by inverting the respective spins at MPS sites $n$ and $n+1$ simultaneously (see Fig. 13):

$$f_\square = f_{\llcorner,n,m} \otimes f_{\lrcorner,n+1,m} + \text{H.c.}, \tag{18}$$

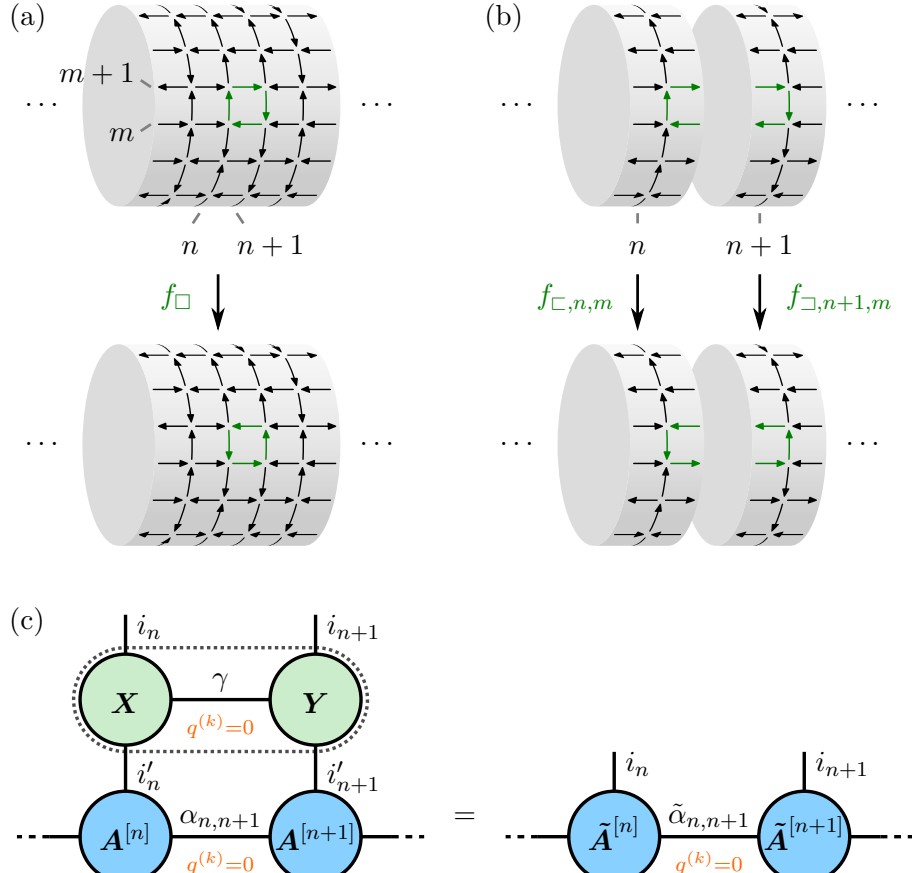

Figure 13: Mapping the Hamiltonian into nearest-neighbour form suitable for TEBD. (a) The plaquette-operator $f_\square$ is applied to an oriented plaquette (green) spanning sites $m, m+1$ and $n, n+1$ on the original lattice. (b) In our computational basis, $f_\square$ factors into nearest-neighbour operators $f_{\llcorner,n,m}$ and $f_{\lrcorner,n+1,m}$, acting separately on MPS sites $n$ and $n+1$. (c) The application of the nearest-neighbour evolution operator $\tilde{U}_{n,n+1}(\tau)$ is made computationally efficient by avoiding the full four-indexed tensor of standard TEBD (dotted line). Instead, two three-indexed tensors $X$, $Y$ (green) are contracted separately into MPS tensors $A^{[n]}$ and $A^{[n+1]}$, followed by a re-compression of the bond-index $\tilde{\alpha}_{n,n+1}$ to the maximal bond-dimension. The gauge-symmetry is reflected by the U(1) invariance of all tensors. In compliance with the constraints on the MPS bonds, the index $\gamma$, which runs over all pairwise interactions, carries quantum number $q^{(k)} = 0$ for $k \equiv n \pmod{2}$.

where $f_{\llcorner,n,m} := \sigma^-_{\boldsymbol{r},n,m+1}\sigma^-_{\boldsymbol{v},n,m}\sigma^+_{\boldsymbol{r},n,m}$ and $f_{\lrcorner,n,m} := \sigma^+_{\boldsymbol{\ell},n,m}\sigma^+_{\boldsymbol{v},n,m}\sigma^-_{\boldsymbol{\ell},n,m+1}$ act concurrently on the redundant horizontal spin-DOFs so as not to violate the constraint Eq. (5). In our notation, $\sigma^\pm_{\boldsymbol{r},n,m}$ raises or lowers the spin on a horizontal link $(n+1/2, m)$ encoded in $|\boldsymbol{r}_n\rangle$. Correspondingly, $\sigma^\pm_{\boldsymbol{\ell},m,n+1}$ acts on the redundant spin of the same link, but encoded in $|\boldsymbol{\ell}_{n+1}\rangle$. Spins on vertical links $(n, m+1/2)$ are encoded in $|\boldsymbol{v}_n\rangle$ and manipulated by $\sigma^\pm_{\boldsymbol{v},n,m}$.

If we apply the above definition to the Hamiltonian of Eq. (1), and group all plaquette operations that apply between sites $n$ and $n+1$ into a single interaction $H_{n,n+1}$, we obtain the nearest-neighbour chain of Sec. 2.4. The interaction consists of $4L_y$ Kronecker products

$$H_{n,n+1} := \sum_{j=1}^{4}\sum_{m=1}^{L_y} h^{(j)}_{\llcorner,n,m} \otimes h^{(j)}_{\lrcorner,n+1,m}, \tag{19}$$

to which the magnetic field terms $-f_\square$ in the Hamiltonian contribute with

$$h^{(1)}_{\llcorner,n,m} = -f_{\llcorner,n,m}\,, \qquad\qquad h^{(1)}_{\lrcorner,n,m} = f_{\lrcorner,n,m}\,, \qquad (20)$$

$$h^{(2)}_{\llcorner,n,m} = -f^\dagger_{\llcorner,n,m}\,, \qquad \text{and} \qquad h^{(2)}_{\lrcorner,n,m} = f^\dagger_{\lrcorner,n,m}\,, \qquad (21)$$

whereas the $\lambda f_\square^2$ part is represented by

$$h^{(3)}_{\llcorner,n,m} = \lambda f^\dagger_{\llcorner,n,m} f_{\llcorner,n,m}\,, \qquad\qquad h^{(3)}_{\lrcorner,n,m} = f^\dagger_{\lrcorner,n,m} f_{\lrcorner,n,m}\,, \qquad (22)$$

$$h^{(4)}_{\llcorner,n,m} = \lambda f_{\llcorner,n,m} f^\dagger_{\llcorner,n,m}\,, \qquad \text{and} \qquad h^{(4)}_{\lrcorner,n,m} = f_{\lrcorner,n,m} f^\dagger_{\lrcorner,n,m}\,. \qquad (23)$$

The TEBD algorithm ultimately requires nearest-neighbour evolution exponentials. However, due to the large local basis, a full expansion of $\exp\{-\tau H_{n,n+1}\}$ into a tensor with four indices (dotted line in Fig. 13(c)) would quickly require impermissibly large memory allocations of 100 GB and more, even if symmetries are exploited. Therefore, we truncate after the first order in the time step $\tau$ and obtain a sum of $4L_y + 1$ products

$$\tilde{U}_{n,n+1}(\tau) := \mathbb{I}_n \otimes \mathbb{I}_{n+1} - \tau H_{n,n+1}\,, \qquad (24)$$

where $\mathbb{I}_n$ denotes the identity operation on MPS site $n$. The advantage of a truncated expansion is that its matrix elements can be expressed as a contraction of two real-valued three-index tensors $X$ and $Y$ over a relatively small virtual index $\gamma$ (see Fig. 13(c)):

$$\langle i_n, i_{n+1}| \tilde{U}_{n,n+1}(\tau) |i'_n, i'_{n+1}\rangle = \sum_{\gamma=1}^{4L_y+1} X^{(\gamma)}_{i_n,i'_n} Y^{(\gamma)}_{i_{n+1},i'_{n+1}}\,. \qquad (25)$$

The application of this operator to an MPS can be performed by separately contracting $X$ and $Y$ into MPS tensors at sites $n$ and $n + 1$, such that computational cost and memory requirements remain manageable despite the large local basis. Furthermore, all involved tensors are symmetry-invariant in accordance with the encoded U(1) quantum numbers, and symmetric tensor contractions preserve the fusion rules of the MPS tensors. In particular, the quantum numbers $q^{(k)}$ associated with bond-indices $\alpha_{n,n+1}$ of an MPS in a fixed gauge-sector must all be zero for $k \equiv n$ (mod 2) (see Sec. 2.3). Owed to the compliance of $\tilde{U}_{n,n+1}(\tau)$ with the redundancy constraint Eq. (5), the same is true for the $\gamma$-index of the interaction. Under the additive U(1) fusion rule, application of $\tilde{U}_{n,n+1}(\tau)$ thus never violates this condition, and (real or imaginary) time-evolution remains restricted to the selected physical sector of the gauge-symmetry.

We continue this discussion in the next section with an overview of local- and bond-dimensions and their impact on the simulation performance.

# D  Performance: Local- and Bond-Dimensions

Our implementation of the TEBD algorithm, as outlined in Sec. 2.4, performs local updates in $O(\chi^3 dK^2) + O(\chi^2 d^2 K)$-time and consumes primary memory of order $O(d^2 K)$, where the factor $K = L_y + 1$ originates from the first-order truncated Taylor-series of the imaginary evolution exponential in Eq. (24). For our linearized two-dimensional systems featuring large local dimensions $d \geq \chi$, this is an essential advantage over standard TEBD, which evolves with full nearest-neighbour exponentials at $O(\chi^3 d^3) + O(\chi^2 d^4)$-time and in our case requires impermissible large amounts of memory, scaling as $O(d^4)$ [31,32]. The lower order evolution errors (at worst $O(\tau^2)$), on the other hand, remain fully under control by extending the total simulation

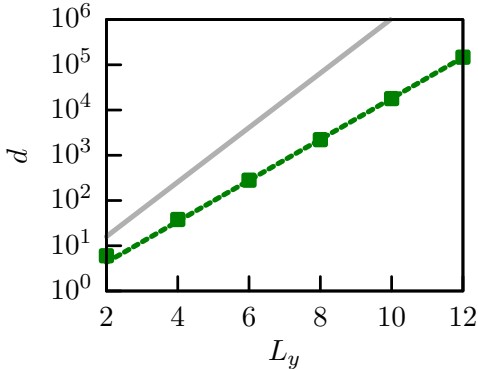

Figure 14: Local MPS dimension (green boxes) over system width $L_y$, in absence of static charges. Compared to unconstrained spin $1/2$ with $\tilde{d} = 4_y^L$ (grey line), the actual dimensions grow with $d \approx 0.53 \times 2.84_y^L$ obtained from log-linear regression (green dashed line).

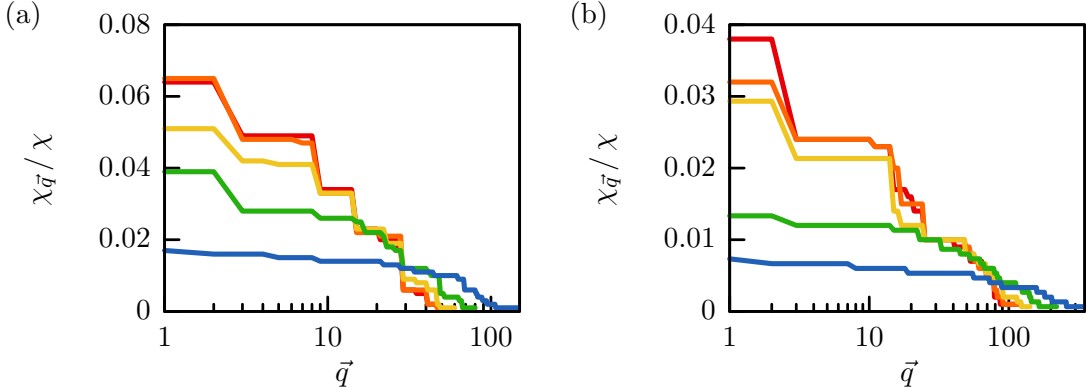

Figure 15: Fractions of the MPS bond-dimension that share equal quantum numbers $\vec{q}$, in descending order, from the final states for $\lambda = -0.7$ (red), $-0.3$ (orange), $0.5$ (yellow), $0.9$ (green) and $0.999$ (blue). (a) Lattice size $18 \times 6$ with $\chi = 1000$, (b) $24 \times 8$ and $\chi = 1500$.

run-time and by actively monitoring convergence against $\Delta E$ while dynamically reducing the time step $\tau$ when necessary. As a consequence, for system sizes where we could still employ both TEBD implementations (up to $L_y = 6$), we obtained ground state approximations with comparable accuracy.

Further performance improvements are enabled by the constraints inherent to the square ice model. Most importantly, the local dimension $d$ of the MPS grows exponentially in the system width $L_y$, and eventually limits our simulations to about $L_y \sim 10$. However, in comparison with $\tilde{d} = 4_y^L$ of an ordinary spin $1/2$ system without constraints, we achieve a reduction to $d \approx 0.53 \times 2.84_y^L$ due to the locally encoded Gauss-law and the selected topological sector $w_x = L_y/2$ (see Fig. 14).

Note that our local growth rate of 2.84 is still larger than Lieb's result $W \approx 1.54$ because the ice rule between adjacent sites $n, n+1$ is not implemented in the local Hilbertspace, but instead imposed through additional constraints mediated by global U(1) symmetries over MPS bonds. Together with the winding number $w_y$, these quantum numbers are carried through the MPS bond-indices and the overall bond-dimension $\chi$ can be decomposed into blocks of indices carrying equal tuples of quantum numbers $\vec{q}$ (which in our case are $q^{(0)}$ and $q^{(1)}$, encoding the

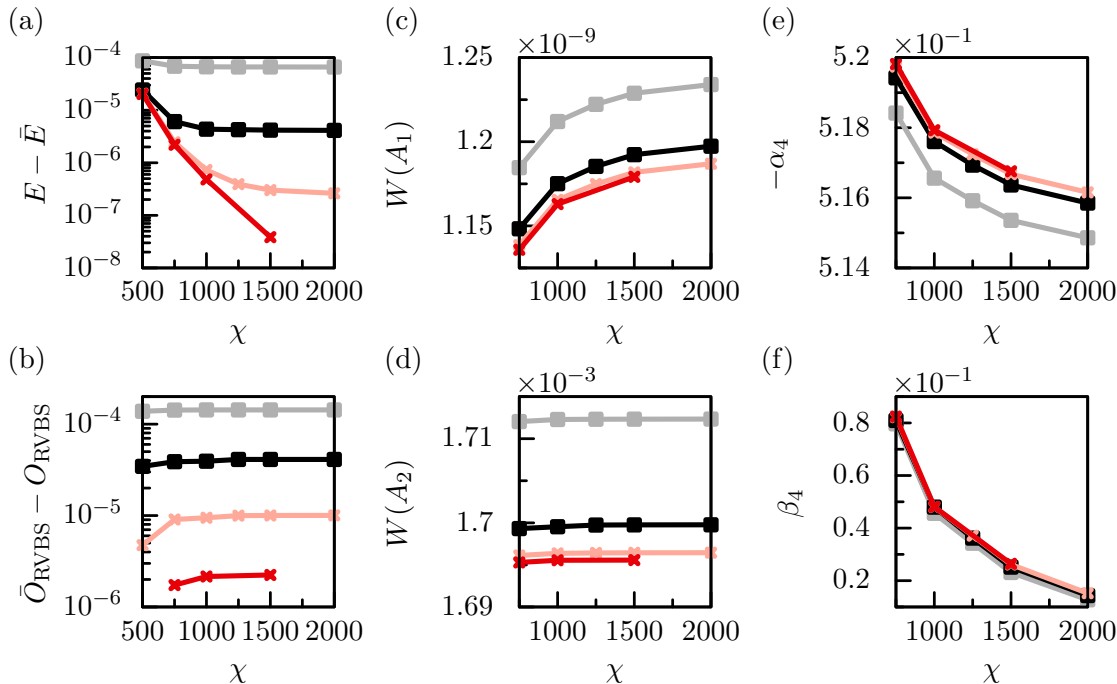

Figure 16: Final state results over MPS bond-dimension $\chi$, from subsequently improved energy convergence thresholds $\Delta E$ (grey and black squares: $10^{-6}$ and $10^{-8}$, rose and red crosses: $10^{-10}$ and $10^{-12}$). Data obtained with an $24 \times 8$ lattice and $\lambda = 0.3$ for (a) Energy and (b) RVBS order, offsetted by extrapolations to $\chi \to \infty$, $\Delta E \to 0$, (c) Wilson loops for rectangles $A_1 \sim 8 \times 4$ and (d) $A_2 \sim 3 \times 2$, (e) Wilson loops fit parameters $\alpha_4$ and (f) $\beta_4$.

spin-configurations on horizontal bonds as detailed in Sec. 2.3, along with a third number $z$ keeping track of $w_y$).

Sizes of these blocks $\chi_{\vec{q}}$ are shown in Fig. 15 in final simulation states for various $\lambda$. While in general, block sizes depend on the current simulation state, we observed that they very quickly freeze out in the final state configuration, in which they remain until the end of the simulation. We additionally found that, within the simulated parameter regime, even the largest block was more than one order of magnitude smaller than the overall dimension $\chi$, and block sizes $\chi_{\vec{q}}$ attain a rather flat distribution when approaching the RK-state at $\lambda \to 1$. This feature immediately reflects the nature of the state at the transition point, which is made of an equal weight superposition of all possible closed loop coverings.

We also point out that because every local basis state carries a different set of quantum numbers (owed to the encoded spin configuration), matrices decompose into blocks with physical dimension $d_{\vec{q}} \equiv 1$. From a performance point of view, we benefit greatly from these small block-sizes, since the dimension $\chi_{\vec{q}}$ determines the actual matrix dimensions in the TEBD linear algebra manipulations. Roughly, memory and runtime complexity reduce by a factor comparable to the relative size of the largest block encountered [47, 48].

# E  Convergence

The precision of our simulation results depends both on the MPS bond-dimension, $\chi$ and the simulation convergence threshold $\Delta E$ which controls the size and number of imaginary evolution time steps (see Sec. 2.4). We have repeated simulations over a wide range in both

parameters in order to check the validity of our simulations and to obtain reliable error estimates. Exemplary results are presented in Fig. 16.

While all results converge asymptotically in both $\chi \to \infty$ and $\Delta E \to 0$, certain order parameters and Wilson loops with small enclosed area are predominantly susceptible to the energy threshold (see, for instance, Figs. 16[(b),(d)]). In those cases, contributions to excited states have not sufficiently been suppressed, and very small time steps are eventually required to cope with the second- (and higher) order errors in the time step. These errors then manifest, for instance, in the form of anisotropies in Wilson-loop measurements due to an unresolved finite-size gap (as discussed in Appendix B). As a counter-measure, we average our measurements over both even and odd sub-lattice positions. We also found that certain simulation results (such as energy and order parameters) can be well approximated by the following two-dimensional function:

$$O(\chi, \Delta E) = \bar{O} + c_1 e^{-|c_2|\chi} + c_3 \Delta E^{-|c_4|}. \tag{26}$$

In situations where our data $O(\chi, \Delta E)$ (typically around 10 data points) was neither converged satisfactorily in $\chi$ nor $\Delta E$, we thus found the unknown extrapolated value $\bar{O}$ among constants $c_1, \ldots, c_4$ (of which $c_1$ and $c_3$ can have different sign, as exemplified by Figs. 16[(b)–(f)]).

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
