# Peer review of "Phase Diagram and Conformal String Excitations of Square Ice using Gauge Invariant Matrix Product States"

_SciPost Physics, doi:SciPost Phys. 6, 028 (2019)_

## Round 1 · Referee Report · Anonymous (Referee 1) · 2018-10-8

Strengths

1. The paper is one of the first ones to report numerical studies, using tensor networks, of lattice gauge theories in more than 1+1d.
2. The authors have applied several study methods and reached similar results.
3. Although not dealing directly with a scenario that would have suffered, had another method been used, from the sign-problem, the methods presented in this work are sign-problem free and hence this work serves as an important benchmark.
4. The paper combines both theory with numerics. Only after the theoretical background is set, the authors turn to the description of numerical methods and discussion of results, keeping them apart.

Weaknesses

1. Some theoretical concepts in the beginning could have benefited from graphical and/or more detailed technical presentation (see below).

Report

I find this work very interesting and important, mostly for the strong points listed above. As written by the authors, the field of tensor network studies of lattice gauge theories, while being relatively new, is growing very fast, offering various new computation methods that allow one to overcome problems encountered by the traditional Monte-Carlo methods of lattice gauge theories. This work presents a very nice and remarkable result for a two dimensional theory with one compact dimension, by converting a cylindrical system into a one dimensional MPS and employing TEBD techniques.
The authors present the model in question (spin ice in two space dimensions) in a very clear manner – both the Hamlitonian and symmetries, and the different phases it exhibits. They further introduce a mapping of the system (when put on a cylinder) to a 1d one, which allows one to construct an MPS ansatz (discussed as well) for the study of the model.
Although this part of the paper (description of the mapping to 1d and construction of the MPS) was very clear to me, I am afraid it might not be clear, in general, to readers not familiar with tensor networks, or lattice gauge theories. In my opinion, subsection 2.3, where the MPS is constructed, can be significantly improved if some figures are added, showing the system (cylinder) and how it is being blocked for the construction of a one dimensional MPS. It can also help if a graphic demonstration of the symmetric MPS construction described in the first paragraph of page 7 ("Here we follow…last MPS bond") is added.
This applies to the next subsection (2.4) – as well. Eq. (8) and the inline equations around it show how to rewrite the Hamiltonian terms in terms of the effective 1d system. A figure and a few more equations (perhaps such equations could be placed in an appendix) can really boost the clarity of discussion.
Then, the authors explain the numerical method and turn, in the following section, to a discussion of the relevant observables and presentation of results. I find these parts of the paper very clearly written, and the results, in my opinion, are clearly presented in the figures given. The combination of "CM-like" observables (section 3.1) and "HEP-like" ones (3.2,3.3) is also very nice and reflects the fact that quantum field theory underlies the same physics, independent of the frame of reference (physical community). In particular, I liked the discussion of relevance of space-space Wilson loops in a model that breaks Lorentz invariance, as the one described in this work, and the nice conclusion about it given at the end of the section (quantitative but not qualitative difference, hence it is still a valid order parameter).
The quantum informative perspective (Entanglement studies in section 4) completes the picture very well.
My last remark connects to the introduction and conclusion sections. The 2d system discussed in the paper has one compact dimension (cylindrical geometry): in such cases, 2d computations are possible by converting the system into an MPS and applying 1 dimension methods, as the authors did. It would be nice to include some comments in the conclusion section about the prospects to performing such calculations in 2d without a compact dimension – what are the possibilities? What are the obstacles? What might have to be changed?
"Conventional" HEP-like lattice gauge theories include fermionic matter, and thus studying them with tensor networks requires the use of fermionic tensor network states (such as fermionic PEPS). Such studies have been carried out by the MPQ collaboration in the last few years, for 2 space dimensions and more (N. J. Phys. 18, 043008 ; Ann. Phys. 363, 385-439 ; Ann. Phys. 374, 84-137 ; Phys. Rev. D 97, 034510). The last one, in particular, suggests a numerical method, combining tensor networks with Monte-Carlo, independent of the space dimension and boundary conditions, which is sign-problem free. I believe that these works should be mentioned along with the other LGT-TN works mentioned in the introduction, as well as in the context of 2d LGT-TN calculations.

Requested changes

1. Not necessary, but potentially useful, as explained in the report – adding figures and equations to sections 2.3 and 2.4 to improve clarity.
2. A comment in the conclusions on general 2d systems and higher dimensions.
3. Referring to the previous works on LGT in 2d (and more) with fermionic PEPS.

  • validity: top
  • significance: high
  • originality: high
  • clarity: high
  • formatting: perfect
  • grammar: perfect

Author:  Ferdinand Tschirsich  on 2019-02-09  [id 433]

(in reply to Report 1 on 2018-10-08)

We thank the referee for her/his report and the suggestions, which we considered in the resubmitted manuscript as follows:

(1)
In subsection 2.3 we have added Fig. 3 to give a pictorial representation of the system, the blocking and the some parts of the symmetric MPS construction. We additionally accompany the paragraph mentioned by the referee with some more general notes in Appendix A.
Following the remarks of the referee on subsection 2.4, we have moved to, and expanded on the details and equations in Appendix C, and added Fig. 13.

(2)
In principle we can use open boundary conditions in the short direction of the MPS, e.g. by explicitly fixing vertical spin-configurations at $m=1/2$ and $m=L_y+1/2$ in the local basis, which in turn also fixes $w_y$. However, the computational cost would be comparable, and we opted for the periodic boundary conditions as to moderate finite-size and boundary effects.

We have added a corresponding sentence at the beginning of Sec. 3, at the end of the first paragraph, where it complements the discussion on aspect ratios:
"While our algorithm is amenable to both periodic- and open boundary conditions along the y-direction at a comparable computational cost, we opted for the cylindrical conditions in all simulations as to minimize finite-size and boundary effects."

(3)
We have cited these works in the introduction together with the other LGT-TN works, as well in our closing remarks in the conclusion right after: "[...] more specialized and powerful and TN classes such as projected entangled pair states".

---

## Round 1 · Referee Report · Anonymous (Referee 2) · 2018-11-8

Strengths

1- analysis of a U(1) link model with MPS 2- interesting characterization of the string excitations in the RVB phase

Weaknesses

1- Use of MPS rather than more powerful TN. 2- Presentation sometimes too technical and with cumbersome notation 3- Exponentially complex trick to get rid of doubling of the Hilbert space 4- The results obtained are not conclusive about several aspects due to the limitation of the ansatz chosen

Report

I have read the paper and found it overall very interesting and well written. Unfortunately the authors have chosen the simplest TN available, the MPS that in this context turns out to be of very limited help, in getting definite answers about relevant quantities such as the string tension and the related Wilson loop area law coefficient. Nevertheless the paper is an interesting addition in the growing field of gauge theories studies with tensor networks.

I first have a side remark. I guess the title is a bit misleading since the present work does not use any of the advanced construction of gauge invariant tensor networks but rather uses a smart trick in which the gauge symmetry is enforced locally on an extended Hilbert space and then the Hilbert space is reduced by using global U(1) symmetries. This is technical but very different from the spirit of using gauge invariant tensor network on the original Hilbert space.

Before recommending it for publication I strongly suggest that the authors take critically in consideration the observations below that should help them improving the presentation and clarifying some controversial aspects of their work.

Detailed report:

1- The notation is very difficult to understand for example s_1 ...s_4 is used to label spins on links below Eq 1 but just the line above the sigma are labelled with 1 to 4. Are they actually acting on the spins s_1 ....? Why not to use the same s_1... to label the sigmas

2- I strongly recommend to define the concept they use, for example what is "pyrochlore lattice" ?

3- The sentence (or equivalent with an Ising model with the same term....) cannot be understood. Which term are they referring to?

4- The sentence at the end of sect 2.2 starting with The nature.... monopole contribution, needs to be reformulated. As it stand mentioning the continuum limit in order to justify confinement is completely misleading. How do the authors plan to construct the continuum limit of the present lattice model? Can they elaborate on this point?

5- Again the notation of Eq 4 is unfortunate, s_i where the spins on the links below Eq 1 and in Fig. 1, adding a bold font does not help the reader, can the authors use another letter of the alphabet?

6- The authors group in a single site all vertical links and the horizontal links on the left and on the right of them. This implies a double counting that is fixed by a new constraint defined in Eq 5. While I would see how to solve the constraint by using a bunch of copy tensors, they decide to go along a different path. My understanding is that they enumerate the configurations of spins on the left and on the right and introduce as many blocks in the tensor as configuration on the horizontal links (an exponential number as a function of the transverse size). This seems to add a lot of overhead to the calculations, have they compared with the more traditional approach based on copy tensors?

7-In any case the explanation would strongly benefit from a schematic drawing of the encoding of their 2D lattice in a 1D TN.

8- In Figure 5 the bond dimension is cited as the source of the error bars, can the authors explain in which sense they are able to associate an error bar with a certain value of the bond dimension and cite the relevant references in which this technique have been used/ tested on known models? I have in mind the recent extrapolations methods based on either the correlation length of the state or the DMRG truncation error.

9- They claim that the critical point is extracted from the finite size scaling, can they be more precise? Is it extracted from the extrapolation of the crossings of appropriately re-scaled curves? Do they need to know a certain critical exponent? Do they use Binder cumulants?

10- The analysis about entanglement does not reflect the current results in the field where entanglement in gauge theories can be divided in two parts distillable entanglement and entanglement due to the symmetry constraints. Can the authors make contact with those results and explain which part of the entanglement they are dealing with?

11- The string entanglement is defined by subtracting the vacuum entropy from the entropy of the string configuration. Is this actually a genuine entropy measure? It could still be that the string state is orthogonal to the ground state but has the same entanglement, why would they then associate zero entanglement to it?

12- The authors seem to apply a standard TEBD algorithm although they mention that the size of the symmetric blocks of the Hamiltonian exceeds 100GB, can they explain how they do this?

13- With respect to the string excitations being described by a free bosonic theory, I find this piece of the work possibly the most interesting one. Is there any understanding of it from the microscopic details of the model? Namely the fact that the strings are extended objects seems to coincide with the fact that the authors obtain a dimensional reduction of the problem from 2D to 1D. Why are the string bosonic, can they show it in terms of braiding and commutation relation? Can they expand a bit this section by adding the relevant discussion in the references they mention?

Requested changes

1- Modify the title, that as it is I find a bit misleading. I would suggest to substitute gauge invariant tensor network with symmetric MPS

Critically address the above comment 1-13

  • validity: good
  • significance: good
  • originality: ok
  • clarity: good
  • formatting: excellent
  • grammar: excellent

Author:  Ferdinand Tschirsich  on 2019-02-09  [id 434]

(in reply to Report 2 on 2018-11-08)

We thank the referee for her/his thoughtful review of the manuscript.

Following the referee's request, we have modified the title by replacing "Gauge Invariant Tensor Networks" with "Gauge Invariant Matrix Product States", which is indeed more specific to the simulations that we performed using MPS confined to the gauge-invariant sector of the spin-ice model.

Based on the referee's initial remark, we have further added a new section in the appendix (Appendix A) in which we make the connection between our gauge-invariant 1D-mapped MPS and the underlying general construction of (abelian and non-abelian) gauge-invariant tensor networks in [5], based on the quantum link model (QLM) formalism. We hope that these additions make it more clear in how far the construction in this manuscript differs from others (especially [6],[19-22], where various types of gauge-invariances have been encoded in tensors with special internal structures). As the referee has pointed out, the QLM construction parameterizes the gauge-invariant sector of the Hilbert space in the local computational basis, but also introduces artificial abelian U(1) local symmetries on the links, which in turn can be encoded with the well-established framework for global abelian symmetries. We find this a very practical and canonical way to reproduce the special internal (block-)structures required for gauge-invariant tensor networks.

Following the referee's detailed remarks, we have carefully reviewed our manuscript, as reported in the following:

(1)
We have clarified our notation as suggested. In the original manuscript, the symbol $s_i$ was used to denote spin variables, the local Hilbert space, as well as individual canonical spin-1/2 basis states on some lattice link $i$. We now use subscripts $\mu$ to refer to links in general, and use the subscripts $\mu_1$, ..., $\mu_4$ to denote the local Hilbert space which the operators acts on. We also introduce $s_\mu$ to label $\sigma^z$ quantum-numbers (and respective eigenstates in bra-ket notaion) in the beginning of subsection 2.3 which are then used throughout the manuscript. In Eq. (4), the bold-faced $\boldsymbol{s}$ (vector) has been replaced by $\boldsymbol{v}$. In context of spin-correlations of Eq. (8), $s$ still appears as a spin coordinate, but only in subscripts.
In the definition of $\omega_y$ at the end of subsection 2.1, we replaced $s$ by $\sigma_z$.
Furthermore, we now denote charges with $\bar{c}$ to distinguish from the $c$ used for central charge and fit constants of the form $c_i$.

(2)
We have rephrased and added a more descriptive wording, the sentence in question reads now:

"[...] on a lattice of corner-sharing cross-linked squares (the two-dimensional analogue of a pyrochlore lattice), in the limit of strong anisotropy $J_{xy}\ll J_z$ [28] (an equivalent mapping can be obtained in Ising models, see Ref. [27])."

The concept is described in more detail in the Refs. [27][28] cited immediately after the paraphrased results.

(3)
The term originally referred to the antiferromagnetic $\sigma^z \sigma^z$ interaction between linked spins, as discussed in Ref. [28] and reviewed in Ref. [27]. We have recast the sentence such that it is now clear it refers to the references.

(4)
The continuum limit of quantum link models can be reached using dimensional reduction, if the theory allows for a Coulomb (deconfined) phase in one dimension more (see the discussion in Phys. Rev. D 60, 094502 (1999), which we have added as Ref. [42] to the reference list). This is indeed applicable to the model we consider.

There is actually one tricky point which deals with Lorentz invariance, that is explicitly broken on the lattice. In particular, the original proof in [43] strictly requires this symmetry to be present, so it is not immediately clear how the conclusions of [43] are applicable to lattice problems at all. This is partly discussed in [27].

We have modified the sentence mentioned by the referee as follows:
"The fact that confinement is the only possible scenario here is related to the continuum-limit behaviour of the theory (recovered via dimensional reduction in quantum link models, see Ref. [42]), where confinement is due to monopole contributions [43]."

(5)
We thank the referee for bringing this ambiguity to our attention. We have resorted to $\epsilon$ to denote singular values.

(6)
We did not perform measurements of the overhead due to quantum-numbers book-keeping. We agree that a comparison with other gauge-invariant tensor network methods would be highly interesting. The referee points out that we must ultimately expect an exponentially scaling number of blocks in the transverse size, which together with the other scalings discussed in Appendix D limits the tractable system size. This is of course expected for any MPS parameterization of such a two-dimensional system due to the necessary growth in bond-dimension.
Generally speaking, a large number of blocks enables our simulations by lowering the memory- and runtime cost of typical blockwise tensor-operation during the TEBD which scale polynomially with the block-dimensions but usually only linear in the number of blocks. A typical block consists of up to several hundred double-precision tensor elements (depending on state, bond-dimension and system size); each block has a single one-dimensional physical index and two bond-indices for which some dimensions can be deferred from Fig. 15.
By using optimized numerical codes for handling abelian symmetries in tensor networks, we expect that overhead added in form of a few integer calculations and indexing operations per block (e.g. additions and inversions from the quantum-number fusion rule; lookups in, and creation of, the internal memory layout holding those blocks) is small in comparison to the resources spent in blockwise tensor-operations. It could however become well noticeable in absolute terms, especially when approaching the RK-point where block-sizes become small.

(7)
We have added Fig. 3, and expanded on some of the details in Appendix C and Fig. 13.

(8)
We have added a paragraph towards the end of subsection 3.1, referring to (now) Fig. 6, including a sentence with more details on how we obtain the respective error bars for the transition points:

"The reported error-bars account for the uncertainty in determining the intersection, which is also limited by simulation errors affecting the value of $O$ (see Appendix E for estimates), which around $\lambda_c$ tend to be dominated by limitations in the bond-dimension."

Since we do not use DMRG but rather a 'Trotter-ized' time-evolution protocol with dynamically scaled time-step, we do not use the truncation errors of the individual time-steps. Instead, as reported in Appendix E, we can fit most of our results against an empirical model Eq. (26) which is sensitive to the bond-dimension and additionally the targeted "precision" (determining the duration of the imaginary time-evolution). Albeit an exploration of the error in terms of correlation length would be interesting, our method appears to be sufficient in the 2D spin-ice setting where finite-size corrections tend to be much more significant than the error due to finite bond-dimension and simulation precision.

(9)
In the paragraph added with our reply (8) in subsection 3.1, we clarify how we extract the phase-transition point from rescaled curves: "In detail, our values reported in Fig. 6 have been found from the intersection of curves [...] rescaled for the different widths $L_y$. To this end, the exponent $\gamma$ was tuned to make all three curves intersect at the closest possible values $\lambda$.".

Given the restrictions in system sizes, and the weak first-order nature of the transition which limits the scaling regime, we do not report extrapolated critical exponents. In light of this, we refer to the parameter $\lambda_c$ at which the phase transition occurs more consistently as 'transition point'.
We decided against using Binder-cumulants, which can be implemented in form of MPO measurements of the higher order correlations but quickly become impractical with the growing (index-)dimensions due to the underlying 2D systems.

(10)
We are dealing with the full entanglement entropy, which is the sum of the two contributions mentioned by the referee. We are actually planning a systematic study of the two separate contributions in the future, as those are immediately accessible with our algorithm (since the reduced density matrices we are dealing with are already in block diagonal form in what is the electric field basis in the Casini-Huerta-Myers approach to the problem).

(11)
This is a point that requires clarification. Our goal here is not to quantify excess entanglement per se, but rather, to understand how entanglement between two regions is affected by the presence of a string connecting them.

This is very hard to do in a fully rigorous manner: the main issue is that inserting charges in the system does necessarily introduce 1) some additional boundary effects, and 2) since the string has a width comparable in size with $L_y$, also some effects due to self-interaction. This is why we resorted to the simple entropy difference $S_{\text{diff}}$: this quantity is clearly defined, convenient to measure, and describes qualitatively (and, in case the points 1 and 2 above can be neglected, quantitatively) the entanglement between two regions given by the string. We interpret this as the string entanglement entropy -- that is, the entropy generated by the presence of a string on the top of the vacuum. This is reminiscent of the string energy discussion in literature.

In case the string state has the same entropy, this would imply that the string itself does not generate any additional entanglement between two subregions. It is possible that, due to extremely strong boundary effects, this actually does not correspond to a correct result. In the case mentioned by the referee, it might be that the entanglement of the string excitation is compensated by some self-interaction effects due to the finite width. We purposely stayed away from regimes where this can happen, by considering only the $\lambda>-0.2$ region. These regimes would require transverse system sizes larger than the one we could access.

(12)
Our statement concerning the impractically large allocations of 100GB and more referred to "standard TEBD", as opposed to the first-order truncated evolution exponentials used in our implementation. We have rephrased our statement and moved it to Appendix C, above Eq. (24), to clarify this point.

(13)
To the best of our knowledge, this relation had only been discussed in 3+1-d gauge theories. In the context of our work, the goal was to show that tensor network simulations offer an alternative angle to this perspective, allowing to tackle it from an entanglement perspective. An understanding of the string in terms of bosonic theory from microscopics is extremely challenging. In the 3+1-d QLM, a discussion has been presented in Ref. [62]. While we would be willing to add this piece of discussion, we prefer not to do so, as the argument presented therein do no immediately transfer down to the 2+1-d case and might generate some confusion. We actually find the 2+1-d problem an intriguing open one in the field, that deserves future work on its own.

About statistics: This shall indeed be possible to check using simulations of a full (quasi-)adiabatic exchange between two strings. Since with tensor networks one has access to the full wave function, quantities related to interference are accessible. However, this presently lies well beyond our computational capabilities.

Following the referee's comment, we have added the following sentence to the conclusions after "....systems despite achieving modest system sizes.":

"From the theoretical viewpoint, our results motivate the search for a microscopic derivation of the bosonic string theory of U(1) quantum link models that could complement the numerical results presented here, and possibly extend those to parameter regimes in the vicinity of the transition point between RVB and Neel phase."

---

## Round 2 · Referee Report · Anonymous (Referee 1) · 2019-2-13

Report

I find that the modifications done by the authors from the previous version are sufficient, and that the comments have been taken into full consideration. In particular, I find the new figures and the explanations now given in appendix C excellent.
Therefore, I find the manuscript suitable for publication.

---

## Round 2 · Referee Report · Anonymous (Referee 2) · 2019-2-15

Report

I appreciated the modifications and the discussion with the authors, hopefully they imoroved the manuscript that is now ready for publication.

Requested changes

None

---

## Round 2 · Author Response

Dear Editor,

thank you for handling our submission. We thank the referees, and appreciate the valuable feedback they provided.

Their reports helped us to substantially improve our manuscript, which we therefore resubmit to SciPost, along with detailed replies to both referees and a list of changes.

Yours sincerely,

Ferdinand Tschirsich, Simone Montangero, and Marcello Dalmonte

Reply to referee report I:

We thank the referee for her/his report and the suggestions, which we considered in the resubmitted manuscript as follows:

(1) In subsection 2.3 we have added Fig. 3 to give a pictorial representation of the system, the blocking and the some parts of the symmetric MPS construction. We additionally accompany the paragraph mentioned by the referee with some more general notes in Appendix A. Following the remarks of the referee on subsection 2.4, we have moved to, and expanded on the details and equations in Appendix C, and added Fig. 13.

(2) In principle we can use open boundary conditions in the short direction of the MPS, e.g. by explicitly fixing vertical spin-configurations at $m=1/2$ and $m=L_y+1/2$ in the local basis, which in turn also fixes $w_y$. However, the computational cost would be comparable, and we opted for the periodic boundary conditions as to moderate finite-size and boundary effects.

We have added a corresponding sentence at the beginning of Sec. 3, at the end of the first paragraph, where it complements the discussion on aspect ratios: "While our algorithm is amenable to both periodic- and open boundary conditions along the y-direction at a comparable computational cost, we opted for the cylindrical conditions in all simulations as to minimize finite-size and boundary effects."

(3) We have cited these works in the introduction together with the other LGT-TN works, as well in our closing remarks in the conclusion right after: "[...] more specialized and powerful and TN classes such as projected entangled pair states".

Reply to referee report II:

We thank the referee for her/his thoughtful review of the manuscript.

Following the referee's request, we have modified the title by replacing "Gauge Invariant Tensor Networks" with "Gauge Invariant Matrix Product States", which is indeed more specific to the simulations that we performed using MPS confined to the gauge-invariant sector of the spin-ice model.

Based on the referee's initial remark, we have further added a new section in the appendix (Appendix A) in which we make the connection between our gauge-invariant 1D-mapped MPS and the underlying general construction of (abelian and non-abelian) gauge-invariant tensor networks in [5], based on the quantum link model (QLM) formalism. We hope that these additions make it more clear in how far the construction in this manuscript differs from others (especially [6],[19-22], where various types of gauge-invariances have been encoded in tensors with special internal structures). As the referee has pointed out, the QLM construction parameterizes the gauge-invariant sector of the Hilbert space in the local computational basis, but also introduces artificial abelian U(1) local symmetries on the links, which in turn can be encoded with the well-established framework for global abelian symmetries. We find this a very practical and canonical way to reproduce the special internal (block-)structures required for gauge-invariant tensor networks.

Following his detailed remarks, we have carefully reviewed our manuscript, as reported in the following:

(1) We have clarified our notation as suggested. In the original manuscript, the symbol $s_i$ was used to denote spin variables, the local Hilbert space, as well as individual canonical spin-1/2 basis states on some lattice link $i$. We now use subscripts $\mu$ to refer to links in general, and use the subscripts $\mu_1$, ..., $\mu_4$ to denote the local Hilbert space which the operators acts on. We also introduce $s_\mu$ to label $\sigma^z$ quantum-numbers (and respective eigenstates in bra-ket notaion) in the beginning of subsection 2.3 which are then used throughout the manuscript. In Eq. (4), the bold-faced $\boldsymbol{s}$ (vector) has been replaced by $\boldsymbol{v}$. In context of spin-correlations of Eq. (8), $s$ still appears as a spin coordinate, but subscripts. In the definition of $\omega_y$ at the end of subsection 2.1, we replaced $s$ by $\sigma_z$. Furthermore, we now denote charges with $\bar{c}$ to distinguish from the $c$ used for central charge and fit constants of the form $c_i$.

(2) We have rephrased and added a more descriptive wording, the sentence in question reads now:

"[...] on a lattice of corner-sharing cross-linked squares (the two-dimensional analogue of a pyrochlore lattice), in the limit of strong anisotropy $J_{xy}\ll J_z$ [28] (an equivalent mapping can be obtained in Ising models, see Ref. [27])."

The concept is described in more detail in the Refs. [27][28] cited immediately after the paraphrased results.

(3) The term originally referred to the antiferromagnetic $\sigma^z \sigma^z$ interaction between linked spins, as discussed in Ref. [28] and reviewed in Ref. [27]. We have recast the sentence such that it is now clear it refers to the references.

(4) The continuum limit of quantum link models can be reached using dimensional reduction, if the theory allows for a Coulomb (deconfined) phase in one dimension more (see the discussion in Phys. Rev. D 60, 094502 (1999), which we have added as Ref. [42] to the reference list). This is indeed applicable to the model we consider.

There is actually one tricky point which deals with Lorentz invariance, that is explicitly broken on the lattice. In particular, the original proof in [43] strictly requires this symmetry to be present, so it is not immediately clear how the conclusions of [43] are applicable to lattice problems at all. This is partly discussed in [27].

We have modified the sentence mentioned by the referee as follows: "The fact that confinement is the only possible scenario here is related to the continuum-limit behaviour of the theory (recovered via dimensional reduction in quantum link models, see Ref. [42]), where confinement is due to monopole contributions [43]."

(5) We thank the referee for bringing this ambiguity to our attention. We have resorted to $\epsilon$ to denote singular values.

(6) We did not perform measurements of the overhead due to quantum-numbers book-keeping. We agree that a comparison with other gauge-invariant tensor network methods would be highly interesting. The referee points out that we must ultimately expect an exponentially scaling number of blocks in the transverse size, which together with the other scalings discussed in Appendix D limits the tractable system size. This is of course expected for any MPS parameterization of such a two-dimensional system due to the necessary growth in bond-dimension. Generally speaking, a large number of blocks enables our simulations by lowering the memory- and runtime cost of typical blockwise tensor-operation during the TEBD which scale polynomially with the block-dimensions but usually only linear in the number of blocks. A typical block consists of up to several hundred double-precision tensor elements (depending on state, bond-dimension and system size); each block has a single one-dimensional physical index and two bond-indices for which some dimensions can be deferred from Fig. 15. By using optimized numerical codes for handling abelian symmetries in tensor networks, we expect that overhead added in form of a few integer calculations and indexing operations per block (e.g. additions and inversions from the quantum-number fusion rule; lookups in, and creation of, the internal memory layout holding those blocks) is small in comparison to the resources spent in blockwise tensor-operations. It could however become well noticeable in absolute terms, especially when approaching the RK-point where block-sizes become small.

(7) We have added Fig. 3, and expanded on some of the details in Appendix C and Fig. 13.

(8) We have added a paragraph towards the end of subsection 3.1, referring to (now) Fig. 6, including a sentence with more details on how we obtain the respective error bars for the transition points:

"The reported error-bars account for the uncertainty in determining the intersection, which is also limited by simulation errors affecting the value of $O$ (see Appendix E for estimates), which around $\lambda_c$ tend to be dominated by limitations in the bond-dimension."

Since we do not use DMRG but rather a 'Trotter-ized' time-evolution protocol with dynamically scaled time-step, we do not use the truncation errors of the individual time-steps. Instead, as reported in Appendix E, we can fit most of our results against an empirical model Eq. (26) which is sensitive to the bond-dimension and additionally the targeted "precision" (determining the duration of the imaginary time-evolution). Albeit an exploration of the error in terms of correlation length would be interesting, our method appears to be sufficient in the 2D spin-ice setting where finite-size corrections tend to be much more significant than the error due to finite bond-dimension and simulation precision.

(9) In the paragraph added with our reply (8) in subsection 3.1, we clarify how we extract the phase-transition point from rescaled curves: "In detail, our values reported in Fig. 6 have been found from the intersection of curves [...] rescaled for the different widths $L_y$. To this end, the exponent $\gamma$ was tuned to make all three curves intersect at the closest possible values $\lambda$.".

Given the restrictions in system sizes, and the weak first-order nature of the transition which limits the scaling regime, we do not report extrapolated critical exponents. In light of this, we refer to the parameter $\lambda_c$ at which the phase transition occurs more consistently as 'transition point'. We decided against using Binder-cumulants, which can be implemented in form of MPO measurements of the higher order correlations but quickly become impractical with the growing (index-)dimensions due to the underlying 2D systems.

(10) We are dealing with the full entanglement entropy, which is the sum of the two contributions mentioned by the referee. We are actually planning a systematic study of the two separate contributions in the future, as those are immediately accessible with our algorithm (since the reduced density matrices we are dealing with are already in block diagonal form in what is the electric field basis in the Casini-Huerta-Myers approach to the problem).

(11) This is a point that requires clarification. Our goal here is not to quantify excess entanglement per se, but rather, to understand how entanglement between two regions is affected by the presence of a string connecting them.

This is very hard to do in a fully rigorous manner: the main issue is that inserting charges in the system does necessarily introduce 1) some additional boundary effects, and 2) since the string has a width comparable in size with $L_y$, also some effects due to self-interaction. This is why we resorted to the simple entropy difference $S_{\text{diff}}$: this quantity is clearly defined, convenient to measure, and describes qualitatively (and, in case the points 1 and 2 above can be neglected, quantitatively) the entanglement between two regions given by the string. We interpret this as the string entanglement entropy -- that is, the entropy generated by the presence of a string on the top of the vacuum. This is reminiscent of the string energy discussion in literature.

In case the string state has the same entropy, this would imply that the string itself does not generate any additional entanglement between two subregions. It is possible that, due to extremely strong boundary effects, this actually does not correspond to a correct result. In the case mentioned by the referee, it might be that the entanglement of the string excitation is compensated by some self-interaction effects due to the finite width. We purposely stayed away from regimes where this can happen, by considering only the $\lambda>-0.2$ region. These regimes would require transverse system sizes larger than the one we could access.

(12) Our statement concerning the impractically large allocations of 100GB and more referred to "standard TEBD", as opposed to the first-order truncated evolution exponentials used in our implementation. We have rephrased our statement and moved it to Appendix C, above Eq. (24), to clarify this point.

(13) To the best of our knowledge, this relation had only been discussed in 3+1-d gauge theories. In the context of our work, the goal was to show that tensor network simulations offer an alternative angle to this perspective, allowing to tackle it from an entanglement perspective. An understanding of the string in terms of bosonic theory from microscopics is extremely challenging. In the 3+1-d QLM, a discussion has been presented in Ref. [62]. While we would be willing to add this piece of discussion, we prefer not to do so, as the argument presented therein do no immediately transfer down to the 2+1-d case and might generate some confusion. We actually find the 2+1-d problem an intriguing open one in the field, that deserves future work on its own.

About statistics: This shall indeed be possible to check using simulations of a full (quasi-)adiabatic exchange between two strings. Since with tensor networks one has access to the full wave function, quantities related to interference are accessible. However, this presently lies well beyond our computational capabilities.

Following the referee's comment, we have added the following sentence to the conclusions after "....systems despite achieving modest system sizes.":

"From the theoretical viewpoint, our results motivate the search for a microscopic derivation of the bosonic string theory of U(1) quantum link models that could complement the numerical results presented here, and possibly extend those to parameter regimes in the vicinity of the transition point between RVB and Neel phase."

---

## Round 2 · List of Changes

• Title changed by replacing "Tensor Networks" with "Matrix Product States"
  • Added Figs. 3 and 13
  • Added appendices A and C
  • Renamed and shortened subsection 2.4, referring to Appendix C
  • Incorporated references on PEPS for LGTs including fermionic matter
  • Modified sentence at the end of subsection 2.2 on continuum limit of quantum link models
  • Expanded on finite-size scaling and error bars of the transition points in Fig. 6
  • Added statement on microscopic derivation of bosonic strings in conclusion
  • Added statement on compact dimension after first paragraph of section 3
  • Changed notions of 'critical point' to 'transition point'
  • Rephrased in Appendix C on memory limitations with fully expanded TEBD evolution exponentials
  • Clarified notation / symbols (spin states, singular values etc.)
  • Minor adjustments of symbols, colors and sizes in Figures
  • Minor changes in wording and spelling fixes
  • Fixed funding information

---

## Editorial Decision

published